# Hyperbolic Neural Population Geometry Benefits Computation

**Dennis Wu** [1 2]   **Yi-Chun Hung** [1]   **Braden Yuille** [3]   **James E. Fitzgerald** [* 4 5]   **Han Liu** [* 1 2 6]

## Abstract

Neural population geometry shapes downstream computation. Recent empirical findings in neurobiology suggest that a hyperbolic structure underlies population activity in the hippocampus. Here we provide a theoretical framework for this phenomenon. First, we propose a plausible construction of hippocampal tuning curves that statistically induces hyperbolic geometry. Next, we establish a connection between neural decoding and associative memory by demonstrating that the Modern Hopfield Network update rule computes the minimum mean-squared-error (MMSE) estimator. Finally, we introduce a novel associative memory model defined in hyperbolic space that yields significantly larger capacity than leading models. Our results suggest that animals encode spatial information as a latent hyperbolic cognitive map, improving both memory capacity and decoding accuracy.

## 1. Introduction

A central goal in understanding the brain is to see how neural activity patterns relate to animal behavior (Kriegeskorte & Wei, 2021). In recent years, advances in large-scale neural recordings have shifted the focus from individual neurons to the collective representations formed by large neural populations. Consequently, there is growing interest in characterizing the neural population geometry induced by these activities (De Kamps et al., 2019; Kriegeskorte & Wei, 2021). This geometric perspective may reveal how animals store and process information (Gallego et al., 2017;

Kriegeskorte & Diedrichsen, 2019; Chung & Abbott, 2021; Mathis et al., 2024).

Similarly, in machine learning (ML), by understanding the latent representation of artificial neural networks (ANNs), we gain insight into how these models learn and store information (Bengio et al., 2013). Recent studies demonstrate how one can borrow insights from population geometry in neural systems to improve ML models (Chung & Abbott, 2021; Chou et al., 2025). A growing number of recent studies suggest that hyperbolic geometry emerges in a range of biological systems (Zhang et al., 2022; Zhou et al., 2018; Lee et al., 2024; Ghaninia et al., 2022). However, these studies remain largely empirical. There has yet to be a theoretical framework that: **(i)** explains how hyperbolic geometry is induced by neural populations, **(ii)** characterizes its impact on downstream decoding, and **(iii)** provides design principles for machine learning models.

For **(i)**, we develop a plausible population coding model of the hippocampus that induces hyperbolic geometry. Specifically, we show that when the widths of Gaussian place fields follow an exponential distribution, the induced semi-metric space is tree-like and statistically hyperbolic in Gromov's sense (Gromov, 1987). Interestingly, this exponential distribution of place field widths matches the experimental observations in Zhang et al. (2023); Rich et al. (2014).

For **(ii)**, we establish a formal link between neural decoding and associative memory. We show that the recall dynamics of modern Hopfield networks (MHNs) approximate the optimal Minimum Mean Square Error (MMSE) estimator. Building on this insight, together with the hyperbolic geometry induced by the neural encoder, we propose a new associative memory model that operates directly in hyperbolic space. Theoretically, we show that this model achieves substantial capacity improvements over previous models (Ramsauer et al., 2020; Krotov & Hopfield, 2021).

Regarding **(iii)**, we demonstrate the utility of this framework for both pattern completion and general machine learning tasks. We show that our hyperbolic associative memory model outperforms existing memory models with superior accuracy on pattern completion. Inspired by the connection between MHNs (Ramsauer et al., 2020) and the attention mechanism (Vaswani et al., 2017), we introduce a hyperbolic memory module that integrates seamlessly into ML

---
[*]Joint senior authors [1]Department of Computer Science, Northwestern University [2]Center for Foundation Models and Generative AI, Northwestern University [3]Integrated Science Program, Northwestern University [4]Departments of Neurobiology, Physics and Astronomy, and Engineering Sciences and Applied Mathematics, Northwestern University [5]NSF-Simons National Institute for Theory and Mathematics in Biology [6]Department of Statistics and Data Science, Northwestern University. Correspondence to: Dennis Wu <hibb@u.northwestern.edu>.

*Proceedings of the 43rd International Conference on Machine Learning*, Seoul, South Korea. PMLR 306, 2026. Copyright 2026 by the author(s).

architectures. Our simulations demonstrate that this model provides substantial performance gains in ML tasks. Notably, these improvements are most pronounced when the hidden dimensionality is constrained, suggesting that hyperbolic geometry offers a more efficient representation space for information storage in low dimensions.

**Organization.** The remainder of this paper is organized as follows. Section 2 introduces the encoding model and derives the Bayes-optimal decoder, which motivates our later connection between neural decoding and associative memory. Section 3 collects the geometric tools we need. The heart of the paper is Section 4, where we show that exponentially-distributed place-field widths induce a statistically hyperbolic semi-metric (Theorem 4.2). Then we build a hyperbolic associative memory model with double-exponential capacity (Theorem 4.8). Section 5 validates these theoretical results through simulations on pattern completion and downstream ML tasks. Section 6 summarizes our findings and implications for computational neuroscience and machine learning. Our limitations can be found in Section A. The overall logic is visualized in Figure 1. The table of notations is in Table 2.

## 2. Neural Computing

This section introduces a neural encoding model based on tuning curves and Poisson spiking, and derives the corresponding Bayes-optimal decoder. We model spatial coding in the hippocampus and denote the space of possible locations by $\mathcal{S}$. A single location is denoted by $s \in \mathcal{S}$. Neural population activity is represented by a vector $n \in \mathbb{R}^N$. The index $i \in [N]$ refers to neurons. We use $\mu \in [M]$ to index latent states or discretized stimulus locations.

### 2.1. Tuning Curve

Neural population codes are commonly modeled using tuning curves (Dayan & Abbott, 2005). Tuning curves are functions describing the firing rate for each neuron when given a stimulus $s$. Each neuron responds preferentially to a subset of the stimulus space. Let $s \in \mathcal{S}$ denote the physical location of the animal. We model the firing rate of a neuron $i$ in the hippocampus by a tuning curve

$$\lambda_i(s) = \sum_{k=1}^{K} \lambda_{ik} \cdot \exp\left(-\frac{||s - s_{i,k}||_2^2}{2\sigma_{ik}^2}\right), \quad (2.1)$$

where $s_{ik}$ denotes the location of the $k$-th place field of neuron $i$, $\sigma_{ik}$ is its width, and $\lambda_{ik}$ is its scale. Neural spiking is modeled as a Poisson process. Given a time window of duration $T$, the spike count $n_i$ for neuron $i$ is modeled as:

$$n_i \mid s \sim \text{Poisson}(\lambda_i(s)T). \quad (2.2)$$

The population response is denoted by $n = (n_1, \cdots, n_N)$.

We will henceforth assume that $K = 1$ and $\lambda_i = \lambda_{\max}$, in which case Equation (2.1) corresponds to the case of uniform spatial tiling through single place fields, as is typical when animals explore a simple environment (Fenton et al., 2008). Future work is needed to explore the multi-field case typical of large environments (Rich et al., 2014). Statistically, one can consider this model as an encoder (generative model) that encodes the input stimulus $s$ into a high-dimensional noisy vector $n$, or into $\lambda(s) = (\lambda_1(s), \cdots, \lambda_N(s))$ without noise. In the next subsection, we review standard statistical approaches to decoding under the Poisson tuning-curve model above, including MLE, MAP, and Bayes-optimal estimators such as the MMSE rule. We then relate these estimators to the update rule of modern Hopfield networks.

### 2.2. From Bayes-Optimal Decoding to Recall

We first explain the standard decoding method for neural tuning curves. Next, we establish an equivalence between the recall dynamics of associative memory models and decoding. Specifically, the intractability of Bayes-optimal decoding naturally motivates a tractable approximation in the form of memory recall. This connection allows us to interpret memory retrieval as a decoding process.

**Statistical Decoders.** From a statistical point of view, neurons encode the stimulus $s$ into a noisy high-dimensional code $n$. Decoding can be formulated as inferring a latent stimulus $s$ from neuronal population activity $n$ under a prescribed loss function. In computational neuroscience (Dayan & Abbott, 2005; Pillow et al., 2005), this process is typically formulated as maximum likelihood estimation (MLE) or maximum a posteriori (MAP) estimation,

$$s_{\text{MLE}}^*(n) = \underset{s}{\arg\max} \log p(n \mid s). \quad (2.3)$$

$$s_{\text{MAP}}^*(n) = \underset{s}{\arg\max} \log p(s \mid n). \quad (2.4)$$

Another common choice is to minimize the expected squared loss, $\ell(\hat{s}) = \mathbb{E}_{p(s|n)}\left[\|s - \hat{s}(n)\|_2^2\right]$. Under this loss, the Bayes-optimal estimator is given by the posterior mean (Hastie, 2009),

$$s_{\text{MMSE}}^*(n) = \int_{\mathcal{S}} p(s \mid n)\, s\, \mathrm{d}s, \quad (2.5)$$

commonly referred to as the Minimum Mean Squared Error (MMSE) estimator.

Although $s_{\text{MAP}}^*(n)$ and $s_{\text{MMSE}}^*(n)$ are not equal in general, the Bernstein–von Mises theorem (Ghosh & Ramamoorthi, 2003) shows that, as the population size $N$ becomes large, the MAP estimate concentrates around the MLE, which is

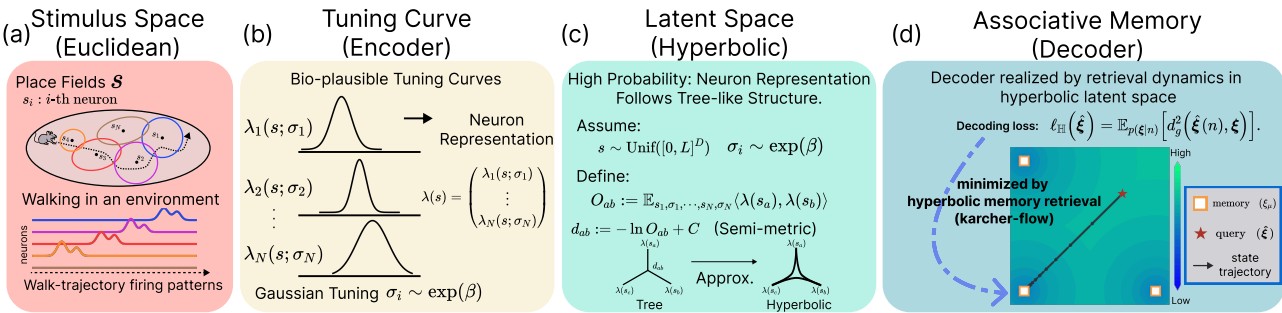

*Figure 1.* **(a)** Illustration of the stimulus space $\mathcal{S}$ and place cell firing patterns. **(b)** Illustration of how neurons encode stimulus $s$ through tuning curves $\lambda$ and output $n(s)$. **(c)** Our constructed tuning curve model that induces hyperbolic geometry. **(d)** Illustration of how the MMSE estimator (decoder) is realized by the memory retrieval dynamics in latent hyperbolic space.

asymptotically normally distributed. Consequently, under regularity conditions, the posterior mean $s^*_{\mathrm{MMSE}}(n)$ converges to $s^*_{\mathrm{MAP}}(n)$. This asymptotic behavior motivates us to link the update rule in an associative memory model to the posterior mean.

To compute $s^*_{\mathrm{MMSE}}$, we discretize the stimulus space into $M$ grid points $\{s_\mu\}_{\mu=1}^M$ and assume a uniform prior, i.e., $p(\mu) = \frac{1}{M}$. Assuming conditional independence of spike counts across neurons given $s$, the posterior takes the form

$$p(s_\mu \mid n) = \frac{e^{h_\mu(n)}}{\sum_\nu e^{h_\nu(n)}} := \mathrm{softmax}_\mu(h_1(n), \cdots, h_M(n)),$$

where $h_\mu(n) = \sum_{i=1}^N n_i \log \lambda_i(s_\mu)$ are log-likelihood scores for each stimulus. The Bayes-optimal decoder (fully derived in Section E.9.2) is therefore

$$s^*(n) = \sum_{\mu=1}^M \mathrm{softmax}_\mu(h(n)) s_\mu. \tag{2.6}$$

**Modern Hopfield Networks.** Modern Hopfield networks (Ramsauer et al., 2020; Krotov & Hopfield, 2021) (MHNs) are associative memory models defined over continuous state spaces. Let $\psi^E : \mathbb{R}^N \to \mathbb{R}^d$ be an embedding map. We define the network state as $v = \psi^E(\lambda(s))$ and stored memory patterns $\xi_\mu = \psi^E(\lambda(s_\mu))$. Given a state $v$, the MHN update is

$$\mathrm{MHN}(v) := \sum_{\mu=1}^M \mathrm{softmax}_\mu\big(h_1^{\mathrm{MHN}}(v), \ldots, h_M^{\mathrm{MHN}}(v)\big)\xi_\mu. \tag{MHN}$$

where $h_\mu^{\mathrm{MHN}}(v) = v^\top \xi_\mu = \langle v, \xi_\mu \rangle$. The shared form of (2.6) and (MHN) implies that MHN is a probabilistic inference machine. As this relation plays a central role in our paper, we formalize it in Section 4.

*Remark* 2.1. Here we introduce a novel notation $\psi^E$. It denotes the non-linear map connecting the tuning curve encoder to MHN, which takes $\lambda(s)$ as input. This decoupling avoids imposing a strict biological constraint between encoder and decoder, while ensuring compatibility with the

Boltzmann form assumed in Proposition 2.2. This mapping is not seen in the typical MHN literature, where existing works do not consider the case where inputs are generated by some encoder (tuning curves).

One can now see that (MHN) and (2.6) are structurally similar. To understand the condition of having MHN to compute the Bayes-optimal estimator, we present the following proposition.

**Proposition 2.2.** *If the posterior takes the Boltzmann form*

$$p(\mu \mid v) = \frac{e^{\langle v, \xi_\mu \rangle}}{\sum_\nu e^{\langle v, \xi_\nu \rangle}}, \tag{2.7}$$

*then the MHN update computes the Bayes-optimal estimator of the following problem*

$$\mathrm{MHN}(v) = \underset{z \in \mathcal{V}}{\mathrm{argmin}} \, \mathbb{E}_{p(\mu|v)}\left[||\xi_\mu - z||_2^2\right]. \tag{2.8}$$

The proof is in Section E.1. Proposition 2.2 establishes a connection between Bayes-optimal decoding and MHN dynamics. Specifically, it demonstrates that under the assumption in (2.7) and the MMSE loss, a single MHN update effectively computes the posterior mean estimator. Proposition 2.2 has a second consequence: associative memory can be viewed as an MMSE estimator whose loss respects the underlying geometry of $\lambda(s)$. This view allows us to derive, in Section 4, a non-Euclidean associative memory model.

## 3. Preliminary on Hyperbolic Geometry

Theorem 4.2 and Theorem 4.8, our main geometric results, require three notions from hyperbolic geometry: $\delta$-hyperbolic metric spaces, the hyperboloid model, and the Fréchet mean. Readers familiar with these can skim to Section 4; we collect them here for self-containedness. For a deeper treatment we refer readers to Jost (2005); Bridson & Haefliger (2013).

**$\delta$-hyperbolic Metric Space.** The following notion measures the extent to which a given metric space resembles a

hyperbolic space in terms of thin triangles. See visualization in the appendix Figure 3.

**Definition 3.1** ($\delta$-thin triangles (Gromov, 1987)). Let $(\mathcal{X}, d)$ be a geodesic metric space.[1] A geodesic triangle $\Delta(x, y, z)$ is a triangle with sides

$$[x, y], \ [y, z], \ [z, x],$$

where $[x, y]$ is the geodesic between $x$ and $y$. We say $\Delta(x, y, z)$ is $\delta$-thin if

$$\forall p \in [x, y], \quad \min \{d(p, [y, z]), \ d(p, [z, x])\} \leq \delta,$$

and cyclic permutations.

**Definition 3.2** ($\delta$-hyperbolic (Gromov, 1987)). A geodesic metric space $(\mathcal{X}, d)$ is $\delta$-hyperbolic if every geodesic triangle in $\mathcal{X}$ is $\delta$-thin. Equivalently for all $x, y, z, w \in \mathcal{X}$,

$$d(x, z) + d(y, w) \leq \tag{3.1}$$
$$\max \{d(x, y) + d(z, w), d(y, z) + d(x, w)\} + 2\delta,$$

which is called the 4-point condition.

In general, low $\delta$ means the geodesic metric space is more tree-like, as tree triangles (Chiswell, 2001) are 0-hyperbolic. For a geodesic metric space to behave like a negatively curved space in Gromov's sense (Gromov, 1987), it requires $\delta$ to be uniform, meaning every geodesic triangle is $\delta$-thin for the same $\delta$, independent of the size of the triangle. Note that any geodesic metric space with finite domain is $\delta$-hyperbolic for some constant $\delta$. Thus, only an infinite domain metric space can fail to be $\delta$-hyperbolic. An indicator that the geodesic metric space is tree-like is that $\delta$ remains $\mathcal{O}(1)$ as the diameter of the domain $L \to \infty$. Next, we introduce a relaxed condition to Definition 3.2.

**Hyperboloid Model.** A $d$-dimensional hyperboloid (Lorentz) model is a Riemannian manifold $(\mathcal{M}^d, g^d)$ equipped with the Riemannian metric tensor $g^d = \text{diag}(-1, 1, \cdots, 1)$ and defined by constant negative curvature $\kappa < 0$, denoted as $\mathbb{H}^d_\kappa$.[2] We will drop the superscript indices on $\mathbb{H}, g, \mathcal{M}$ whenever $d$ is clear from the context or irrelevant.

Each point $\boldsymbol{x} \in \mathbb{H}^d_\kappa$ has the parameterized form $[\boldsymbol{x}_t, \boldsymbol{x}_s]$, where $\boldsymbol{x}_t \in \mathbb{R}$ and $\boldsymbol{x}_s \in \mathbb{R}^d$. $\mathbb{H}^d_\kappa$ is equipped with the Lorentz (Minkowski) inner product. For points $\boldsymbol{x}, \boldsymbol{y} \in \mathbb{H}^d_\kappa$, their inner product $\langle \boldsymbol{x}, \boldsymbol{y} \rangle_L$ is given by

$$\langle \boldsymbol{x}, \boldsymbol{y} \rangle_L = -\boldsymbol{x}_t \boldsymbol{y}_t + \boldsymbol{x}_s^\top \boldsymbol{y}_s = \boldsymbol{x}^\top g^d_\kappa \boldsymbol{y},$$

with $\|\boldsymbol{x}\|_L := \sqrt{|\langle \boldsymbol{x}, \boldsymbol{x} \rangle_L|}$ being the Lorentzian norm. Formally, $\mathcal{M}^d$ is the set

$$\mathcal{M}^d = \left\{ \boldsymbol{x} \in \mathbb{R}^{d+1} : \langle \boldsymbol{x}, \boldsymbol{x} \rangle_L = \frac{1}{\kappa}, \boldsymbol{x}_t > 0 \right\}.$$

---

[1] A metric space $(\mathcal{X}, d)$ is geodesic if there exists a geodesic between any two points $x, y \in \mathcal{X}$.

[2] This work considers simply connected Hadamard manifolds.

The origin $\boldsymbol{o} \in \mathbb{H}^d_\kappa$ is the point $[\sqrt{-1/\kappa}, 0, \cdots, 0]^\top$.

We denote the distance of any two points $\boldsymbol{x}, \boldsymbol{y} \in \mathcal{M}$ as $d_g(\boldsymbol{x}, \boldsymbol{y})$. Notably, in the hyperboloid model, hyperbolic distance can be expressed in terms of the Lorentz inner product as $\cosh(|\kappa|^{1/2} d_g(\boldsymbol{x}, \boldsymbol{y})) = -\kappa \langle \boldsymbol{x}, \boldsymbol{y} \rangle_L$.

**Tangent Space.** The tangent space at a point $\boldsymbol{x} \in \mathbb{H}^d_\kappa$ is the set of points orthogonal to $\boldsymbol{x}$, defined as

$$T_{\boldsymbol{x}} \mathbb{H}^d_\kappa = \left\{ \boldsymbol{y} \in \mathbb{R}^{d+1} : \langle \boldsymbol{x}, \boldsymbol{y} \rangle_L = 0 \right\}, \tag{3.2}$$

where the tangent space is isometric to the Euclidean space (Lee, 2018). If a manifold is smooth, it admits a tangent space at every point. For any $\boldsymbol{x}, \boldsymbol{y} \in \mathbb{H}^d_\kappa$ and $\mathrm{v} \in T_{\boldsymbol{x}} \mathbb{H}^d_\kappa$, the exponential map $\text{Exp}_{\boldsymbol{x}}(\cdot) : T_{\boldsymbol{x}} \mathbb{H}^d_\kappa \mapsto \mathbb{H}^d_\kappa$, and the inverse exponential map $\text{Exp}^{-1}_{\boldsymbol{x}} : \mathbb{H}^d_\kappa \mapsto T_{\boldsymbol{x}} \mathbb{H}^d_\kappa$ are given by

$$\begin{cases} \text{Exp}_{\boldsymbol{x}}(\mathrm{v}) & = \cosh(\sqrt{-\kappa} \|\mathrm{v}\|_L) \boldsymbol{x} + \frac{\sinh(\sqrt{-\kappa} \|\mathrm{v}\|_L)}{\sqrt{-\kappa} \|\mathrm{v}\|_L} \mathrm{v}, \\ \text{Exp}^{-1}_{\boldsymbol{x}}(\boldsymbol{y}) & = \frac{\cosh^{-1}(-\kappa \langle \boldsymbol{x}, \boldsymbol{y} \rangle_L)}{\sqrt{(-\kappa \langle \boldsymbol{x}, \boldsymbol{y} \rangle_L)^2 - 1}} \left( \boldsymbol{y} + \kappa \langle \boldsymbol{x}, \boldsymbol{y} \rangle_L \boldsymbol{x} \right). \end{cases}$$

For a smooth manifold $\mathcal{M}$ and a point $p \in \mathcal{M}$, the tangent space $T_p \mathcal{M}$ is the vector space that represents the first-order approximation of $\mathcal{M}$ at $p$, which then describes the neighborhood of $p$ to first order. $T_p \mathcal{M}$ consists of all possible velocities of smooth curves through $p$. To relate this linear structure back to the manifold, the exponential map sends each tangent vector $\mathrm{v}$ to the endpoint of the geodesic starting at $p$ with initial velocity $v$, evaluated at unit time. This map allows operations defined linearly on $T_p \mathcal{M}$ to be transported to $\mathcal{M}$ itself.

**Fréchet mean.** Next, we define the Fréchet mean, which computes the geometric mean under general metrics.

**Definition 3.3** (Fréchet mean). Given a set of points $\mathcal{B} = \{\boldsymbol{x}_\mu\}_{\mu=1}^M$ on a Riemannian manifold $(\mathcal{M}, g)$. The Fréchet mean of them, denoted as $\text{FM}(\mathcal{B})$, is defined as the solution and optimal values of the following optimization problem (Bačák, 2014)

$$\text{FM}(\mathcal{B}) := \underset{\boldsymbol{z} \in \mathcal{M}}{\text{argmin}} \frac{1}{M} \sum_{\mu=1}^M d_g^2(\boldsymbol{x}_\mu, \boldsymbol{z}).$$

Furthermore, given a set of weights $\mathcal{W} = \{w_1, \cdots, w_M\} \in \mathbb{R}^+$, the weighted Fréchet mean $\text{WFM}(\mathcal{B}, \mathcal{W})$ over $\mathcal{B}$ is the solution to the following optimization problem

$$\text{WFM}(\mathcal{B}, \mathcal{W}) := \underset{\boldsymbol{z} \in \mathcal{M}}{\text{argmin}} \sum_{\mu=1}^M w_\mu \cdot d_g^2(\boldsymbol{x}_\mu, \boldsymbol{z}).$$

## 4. Hyperbolic Neural Population Geometry

In this section, we introduce a plausible tuning curve model that approximates hyperbolic geometry via neural population codes. Our setup is inspired by the representational

similarity analysis (RSA) (Kriegeskorte et al., 2008). We look at the pair-wise distance of neural activity patterns between different stimuli, and then study the semi-metric space[3] that underlies it. Next, we discuss the corresponding decoder in hyperbolic geometry and propose a plausible hyperbolic associative memory model with large capacity.

## 4.1. Geometry of Neural Population Activities

Recent findings in neurobiology (Zhang et al., 2023) show that in the CA1 region of the hippocampus, the place field sizes of the tuning curves $\sigma_{i\mu}$ are well approximated by an exponential distribution. Zhang et al. (2023) discover that this distribution is in turn well approximated by uniform sampling in a hyperbolic ball,

$$p(\sigma) \approx \zeta \cdot e^{-\zeta\sigma} \approx \zeta \frac{\sinh(\zeta(\sigma_{\max} - \sigma))}{\cosh(\zeta\sigma_{\max}) - 1}, \qquad (4.1)$$

where $\sigma$ equals the distance from the origin $o$, $\sigma_{\max} \geq \sigma > 0$, and $\zeta = (N - 1)\sqrt{|\kappa|}$, with $\kappa < 0$ and $N > 1$ as the curvature and dimensionality of the space.

Biologically, this corresponds to: (a) most hippocampal place cells have small, localized place fields; and (b) a smaller number of neurons have progressively larger fields. This implies that population activity is sparse and localized, as most neurons respond only in a small region around their preferred location. In our next analysis, we will show that not only is the representation sparse, but the induced stimulus distance is also tree-like.

**Stimulus Distance in Population Codes.** Here we study the geometry induced by the exponential place-field size distribution under (2.1). Define the population response vector given stimulus $s$ (assume noiseless),

$$\lambda(s) = (\lambda_1(s), \cdots, \lambda_N(s)) \in \mathbb{R}^N.$$

Let the stimulus space be $\mathcal{S} = [0, L]^D$, equipped with the Euclidean metric. Assuming a single place field with fixed amplitude, the Gaussian tuning curve of neuron $i$ is

$$\lambda_i(s) = \lambda_{\max} \exp\left(-\frac{||s - s_i||_2^2}{2\sigma_i^2}\right), \sigma_i \sim \mathrm{Exp}(\beta),$$

where centers $\{s_i\}_{i=1}^N$ are distributed according to a Poisson point process with density $\rho$ (Rich et al., 2014), and $\beta$ is the mean of the exponentially distributed place field sizes. We measure distances between population responses $\lambda(s) \in \mathbb{R}^N$. Assuming $\min_i \sigma_i > 0$, we define a semi-metric between stimuli as

$$d_{ab} = -\phi(\langle\lambda(s_a), \lambda(s_b)\rangle) + C, \quad s_a, s_b \in \mathcal{S},$$

---
[3]The space is semi-metric because the triangle inequality is not satisfied.

where $\phi$ is any increasing and continuous scalar function, and $C$ is an universal constant depending only on $\lambda_{\max}$, $N$ and the Lipschitz constant of $\phi$. We define stimulus distances, $d_{ab}$, to be functions of firing-rate inner products to match the standard RSA assumption (Kriegeskorte et al., 2008). This makes our definition consistent with many other neuroscience studies. We mainly study the case where $\phi = \ln$. This is for the simplicity of the proof; one can obtain a similar result using the Lipschitz constant of $\phi$.

Because the widths $\sigma_i$ are random variables, $d_{ab}$ is a *semi-metric* rather than a metric. Concretely, for any three stimuli $s_a, s_b, s_c$, there exist realizations of $\{\sigma_i\}_{i=1}^N$ under which $d_{ab} + d_{bc} < d_{ac}$, violating the triangle inequality. In order to gain insight from the hyperbolic structure of the neural population code, we introduce a definition that allows us to study the hyperbolicity of $d_{ab}$.

**Definition 4.1** (Statistically $\delta$-hyperbolic). Let $\mathcal{S}$ be a space equipped with a distance function $d$. We say $(\mathcal{S}, d)$ is *statistically $\delta$-hyperbolic* at confidence $1 - \eta$ if there exists a constant $\delta > 0$ such that

$$\Pr\left[\Delta(s_x, s_y, s_z, s_w) > 2\delta\right] < \eta,$$

where $s_x, s_y, s_z, s_w$ are sampled independently and uniformly from $\mathcal{S}$, and $\Delta$ denotes the empirical Gromov four-point excess in (3.1).

Unlike Definition 3.2, we relax the universal 4-point condition to a probabilistic one. Rather than requiring the 4-point condition to hold for *every* choice of the quadruples, we only require it to hold for a $(1 - \eta)$-fraction of quadruples drawn uniformly at random. This relaxation is motivated by the case where one needs to study neural population geometry under some observed overall statistics of the neurons.

For any quadruple $(s_a, s_b, s_c, s_d)$ uniformly i.i.d. sampled from $\mathcal{S}$, let $L_{(1)} \geq L_{(2)} \geq L_{(3)}$ be the sorted values of the pair-sum set $\mathcal{P} := \{d_{ab} + d_{cd}, d_{ac} + d_{bd}, d_{ad} + d_{bc}\}$. For convenience, we'll write $\max_{(k)} A$ for the $k$-th largest member of $A$:

$$L_{(1)} = \max{}_{(1)}\mathcal{P}, \quad L_{(2)} = \max{}_{(2)}\mathcal{P}, \quad L_{(3)} = \max{}_{(3)}\mathcal{P}.$$

Following Definition 4.1, we define $\Delta$ as the difference between the two largest sums of opposite distances $\Delta(a, b, c, d) := L_{(1)} - L_{(2)}$. Recall that from Definition 4.1, a semi-metric space is said to be statistically $\delta$-hyperbolic if under uniform sampling in $\mathcal{S}$, $\Delta \leq 2\delta$ with high probability. Now we are ready to present our result on the induced inter-stimulus geometry.

**Theorem 4.2.** *Let $s_a, s_b, s_c, s_d \in \mathcal{S}$ be any quadruple each independently sampled uniformly from $\mathcal{S}$. For any $\eta > 0$ and $N = \mathcal{O}((L/\beta)^D)$, there exists a constant $\delta(\beta, \rho)$ such that*

$$Pr[\Delta(s_a, s_b, s_c, s_d) > 2\delta] < \eta.$$

*Furthermore, $\delta$ is non-trivial since $\lim_{L \to \infty} \frac{\delta}{L} = 0$.*

*Proof Sketch.* We compute the 4-point excess in Definition 3.2 to show that $d_{ab}$ is statistically hyperbolic. The proof is given in Section E.2, and the discussion is provided in Section E.2.1. Note that one can also obtain a similar bound by replacing $\ln$ in $d_{ab}$ with any other increasing and continuous function that is Lipschitz. We choose the specific function $\ln$ for simplicity of the proof. $\square$

*Remark* 4.3. This theorem implies with a sufficient number of neurons $N$, the distance induced by neural population activity $\lambda(s) \in \mathbb{R}^N$ is tree-like. This structure arises from incorporating an exponential place-field size distribution into Gaussian tuning. This corresponds to the hippocampus encoding stimulus distances using a hyperbolic semi-metric. This is analogous to neurons representing spatial information as latent hyperbolic embedding. Not only is this result inspired by experimental findings, but it also provides a realizable construction of tree-like geometry induced by neural populations. Numerical simulations validating Theorem 4.2 can be found in Section G.2 and its Figure 5.

## 4.2. Hyperbolic Decoder

Theorem 4.2 establishes that population activity induces a hyperbolic semi-metric on the stimuli. To mirror the construction in Section 2.2, where we moved from the MMSE estimator to an associative memory model, we commit to a Riemannian manifold that is intrinsically $\delta$-hyperbolic: the hyperboloid (Lorentz) model $\mathbb{H}_\kappa^d$. This model has two properties we rely on. First, it is intrinsically $\delta$-hyperbolic. Second, its Lorentz inner product encodes geodesic distance through a single linear operation, a property we exploit heavily in Section 4.3. Together, these properties let us define the memory model as the MMSE estimator under geodesic rather than Euclidean loss.

Given a set of preferred stimuli $\{s_\mu\}_{\mu=1}^M$, assume there exists a function $\psi^H : \mathbb{R}^N \to \mathbb{H}_\kappa^d$ such that $\boldsymbol{\xi}_\mu = \psi^H(\lambda(s_\mu))$ [4]. Under this latent model, information about $s$ may be inferred by estimating its latent representation $\boldsymbol{\xi}$. The observed input query is $\boldsymbol{\xi}(s) = \psi^H(\lambda(s))$ generated from the firing-rate vector. Decoding is formulated as a squared-loss estimation problem in $\mathbb{H}$. The expected squared geodesic loss and the optimal estimator under it are defined as following (Pennec, 2006):

$$\ell_{\mathbb{H}}\left(\hat{\boldsymbol{\xi}}\right) = \mathbb{E}_{p(\boldsymbol{\xi}|n)}\left[d_g^2\left(\hat{\boldsymbol{\xi}}(n), \boldsymbol{\xi}\right)\right]. \quad (4.2)$$

$$\boldsymbol{\xi}_{\text{MMSE}}^*(n) = \underset{\boldsymbol{z}}{\arg\min} \int_{\boldsymbol{\xi} \in \mathbb{H}_\kappa^d} p(\boldsymbol{\xi} \mid n) \cdot d_g^2(\boldsymbol{z}, \boldsymbol{\xi}) d\boldsymbol{\xi}, \quad (4.3)$$

[4]Here, we again assume the existence of a mapping between the neural encoder and decoder, as in Section 2.

where (4.3) is considered as the posterior Fréchet mean.

Similar to Section 2.2, If we again discretize the domain $\mathcal{S}$ into $M$ points $\{s_\mu\}_{\mu=1}^M$, the optimal decoder under the same loss has the form:

$$\boldsymbol{\xi}^*(n) = \text{WFM}\left(\{\boldsymbol{\xi}_\mu\}, \{p(\mu \mid n)\}\right), \quad \mu \in [M], \quad (4.4)$$

where the posterior over memory indices, $p(\mu \mid n)$, is discussed and defined in hyperbolic space in Section 4.3.

**From Fréchet mean to Karcher flow.** The above optimal estimator is generally not available in closed form. A common way to compute (4.4) is to apply the Karcher flow algorithm (Karcher, 1977). Given some iterate $\boldsymbol{\xi}^t$ at iteration $t$, it approximates (4.4) by iterating

$$\boldsymbol{\xi}^{t+1} = \text{Exp}_{\boldsymbol{\xi}^t}\left(\sum_{\mu=1}^M p(\mu \mid n) \cdot \text{Exp}_{\boldsymbol{\xi}^t}^{-1}(\boldsymbol{\xi}_\mu)\right). \quad (4.5)$$

Here the weights are the posterior, and the average is performed in the tangent space $T_{\boldsymbol{\xi}^t}\mathbb{H}$. Karcher flow not only makes the decoder tractable, but it also approximates $\boldsymbol{\xi}^*$ via iterative updates and converges to the weighted frechet mean (Karcher, 1977). As we show in the next subsection, this corresponds to the recurrent update in associative memory.

## 4.3. Associative Memory in Hyperbolic Geometry

Inspired by the connection between MHN and neural tuning curves, we now define the following update rule for the hyperbolic memory model.

**Definition 4.4** (Karcher-flow model). Given memory patterns $\{\boldsymbol{\xi}_\mu\}_{\mu=1}^M \subseteq \mathbb{H}_\kappa^d$, and some input query $\boldsymbol{v} \in \mathbb{H}_\kappa^d$. The Karcher-flow model update $H : \mathbb{H}_\kappa^d \to \mathbb{H}_\kappa^d$ is defined by

$$H(\boldsymbol{v}) := \text{Exp}_{\boldsymbol{v}}\left(\sum_{\mu=1}^M w_\mu(\boldsymbol{v}) \cdot \text{Exp}_{\boldsymbol{v}}^{-1}(\boldsymbol{\xi}_\mu)\right), \quad \text{(KFM)}$$

where $w_\mu(\boldsymbol{v}) = \frac{e^{\langle \boldsymbol{v}, \boldsymbol{\xi}_\mu \rangle_L}}{\sum_\nu e^{\langle \boldsymbol{v}, \boldsymbol{\xi}_\nu \rangle_L}}$.

Definition 4.4 has two major distinctions from the MHN. First, (KFM) is defined in $\mathbb{H}_\kappa^d$. Second, we use the Lorentz inner product instead of Euclidean inner product. This provides an advantage to KFM: the Lorentz inner product naturally encodes hyperbolic distance. In contrast, the Euclidean inner product in general does not represent Euclidean distance well. This allows our model to better distinguish patterns with similar angular directions but different norms with minimal cost, as $\langle \cdot, \cdot \rangle_L$ has the same complexity as $\langle \cdot, \cdot \rangle$. Furthermore, the form of $\langle \cdot, \cdot \rangle_L$ computationally has the same structure as the $d_{ab}$ used in Theorem 4.2.

Next, we introduce a similar setup to that of Proposition 2.2. Let the latent index be distributed, $\mu \sim p(\mu)$, and denote the

random query in hyperbolic space as $\boldsymbol{v}$. Now we are ready to present the hyperbolic correspondence to Proposition 2.2.

**Proposition 4.5.** *If we model the posterior over indices by the Boltzmann distribution*

$$p(\mu \mid \boldsymbol{v}) = \frac{e^{\beta \langle \boldsymbol{v}, \boldsymbol{\xi}_\mu \rangle_L}}{\sum_{\nu=1}^M e^{\beta \langle \boldsymbol{v}, \boldsymbol{\xi}_\nu \rangle_L}}, \qquad (4.6)$$

*and*

$$\langle \boldsymbol{v}, \boldsymbol{\xi}_\mu \rangle_L = \log p(\boldsymbol{v} \mid \mu) + \log p(\mu) + C(\boldsymbol{v}). \qquad (4.7)$$

*Then* (KFM) *approximates the following estimator via the Karcher flow*

$$H(\boldsymbol{v}) \approx \xi^*(\boldsymbol{v}) = \operatorname*{argmin}_{z \in \mathcal{M}} \mathbb{E}_{p(\mu|v)} \big[ d_{\mathcal{M}}^2(\boldsymbol{\xi}_\mu, z) \big]. \qquad (4.8)$$

*Remark* 4.6. One can consider the above proposition as the non-Euclidean version of Proposition 2.2. However, different from Proposition 2.2, our tractable update rule (KFM) is not the exact optimal estimator. Instead, to converge to the optimal Fréchet mean estimator, one should consider fixed weights throughout the iterative updates. Given some initial state $\boldsymbol{v}^{(0)}$, we set $w_\mu(\boldsymbol{v}) = w_\mu(\boldsymbol{v}^{(0)})$. This reduces the computational cost for $w_\mu(\boldsymbol{v})$. However, in our later analysis, both update rules have the same order of stable fixed points under pattern separation (Theorem 4.8). In the context of modern Hopfield networks, one can think of (4.8) as approximate minimization of the loss of the system, $\ell_{\mathbb{H}}(\hat{\boldsymbol{\xi}})$, via memory retrieval dynamics given by (KFM) .

### 4.4. Pattern Completion

Here we study the problem of pattern completion under $\mathbb{H}_\kappa^d$. Informally, we show that the Karcher-flow model has capacity that scales exponentially in $d$, the dimensionality of the hyperbolic embedding, and doubly exponential in $r_{\max}$, the largest norm among the memory patterns. This capacity improves the capacity of the MHN by a rate of double exponential in $r_{\max}$. We identify the Euclidean space $\mathbb{R}^d$ with the tangent space at the origin, $T_{\boldsymbol{o}} \mathbb{H}_\kappa^d$.

Let $\{x_\mu\}_{\mu=1}^M \subseteq T_{\boldsymbol{o}} \mathbb{H}_\kappa^d$ be the stored patterns. We assume that the pattern embeddings satisfy $\boldsymbol{\xi}_\mu = \operatorname{Exp}_{\boldsymbol{o}}(x_\mu)$, for $\mu \in [M]$. A query is generated by first sampling $\mu \sim \operatorname{Unif}([M])$. Conditioned on $\mu$, we sample the query $\boldsymbol{v}$ via

$$\boldsymbol{v} = \operatorname{Exp}_{\boldsymbol{\xi}_\mu}(\mathrm{v}), \qquad \mathrm{v} = x_\mu + \boldsymbol{\sigma} z, \qquad z \sim \mathcal{N}(0, \boldsymbol{I}_d).$$

Given a decoder $H : \mathbb{H}_\kappa^d \to \mathbb{H}_\kappa^d$ and a target tolerance $\varepsilon > 0$, we aim to control the recall success probability

$$P_{\mathrm{rec}}(\varepsilon) := \Pr \big[ d_{\mathbb{H}}^2 \big( H(\boldsymbol{v}), \boldsymbol{\xi}_\mu \big) \le \varepsilon \big],$$

where the probability is taken over $\mu$ and the query noise. Note that the above condition matches the calculations used to define the memory capacity in classic work (Hopfield, 1982; 1984).

**Assumption 4.7** (Chernoff condition). Fix distinct $\mu \ne \nu$. Under the conditional law $\mathrm{v} \mid \mu$, define

$$\Delta_{\mu\nu}^{\mathrm{E}}(\mathrm{v}) := \|\mathrm{v} - s_\mu\|_2^2 - \|\mathrm{v} - s_\nu\|_2^2.$$

Assume there exist constants $\gamma_{\mathrm{E}} > 0$, $K_{\mathrm{E}} < \infty$, and $d_0 \in \mathbb{N}$ such that for all $d \ge d_0$:

(A1)  $\mathbb{E}[\Delta_{\mu\nu}^{\mathrm{E}}(\mathrm{v}) \mid \mu] \ge \gamma_{\mathrm{E}} d$.

(A2)  Conditioned on $\mu$, $\Delta_{\mu\nu}^{\mathrm{E}}(\mathrm{v}) - \mathbb{E}[\Delta_{\mu\nu}^{\mathrm{E}}(\mathrm{v}) \mid \mu]$ is sub-Gaussian with proxy variance at most $K_{\mathrm{E}} \boldsymbol{\sigma}^2 d$.

Note that both assumptions are naturally satisified when the memory patterns are uniformly distributed and noise is added as above.

**Theorem 4.8.** *Let $\kappa < 0$ and $\alpha = \sqrt{|\kappa|}$. Under Assumption 4.7, if both*

$$\boldsymbol{\sigma} = O\left( r_{\min} \cdot \min\left\{ \frac{1}{\sqrt{d}}, \ \sqrt{|\kappa|} \, e^{-\alpha r_{\min}} \right\} \right)$$

*and*

$$r_{\max} - r_{\min} = o\left( \frac{d}{\alpha r_{\min}^2} \right), \quad \log M = \Theta\left( \frac{d}{|\kappa|} \frac{e^{2\alpha r_{\min}}}{r_{\min}^2} \right)$$

*hold. Then as $d \to \infty$,*

$$\lim_{d \to \infty} P_{rec}(\varepsilon) = 1.$$

The proof can be found in Section E.3. Intuitively, this says that memory retrieval is successful with high probability within a nonzero radius basin of attraction as long as the memory patterns have sufficient separation in their amplitudes. The capacity for different regimes of $\Delta_r$ is summarized in Table 3.

*Remark* 4.9. A notable difference between our assumptions and prior work (Ramsauer et al., 2020; Santos et al., 2024) is that we do not require memory patterns to be normalized. This relaxation is possible because the Lorentz inner product encodes $d_g(\cdot, \cdot)$ in $\mathbb{H}^d$, rather than angular similarity. This provides significant benefits for storing memories in hyperbolic space: the Lorentz inner product has the same computational complexity as Euclidean inner products, yet captures the nonlinear quantity $e^{d_g(\boldsymbol{a}, \boldsymbol{b})}$. Furthermore, distinguishing patterns via the Lorentz inner product rather than the Euclidean inner product allows models to separate patterns with similar directions but different magnitudes.

## 5. Simulations

We conduct simulations on pattern completion, image classification, and multiple instance learning. We compare our model with Dense Associative Memory (DAM) (Krotov &

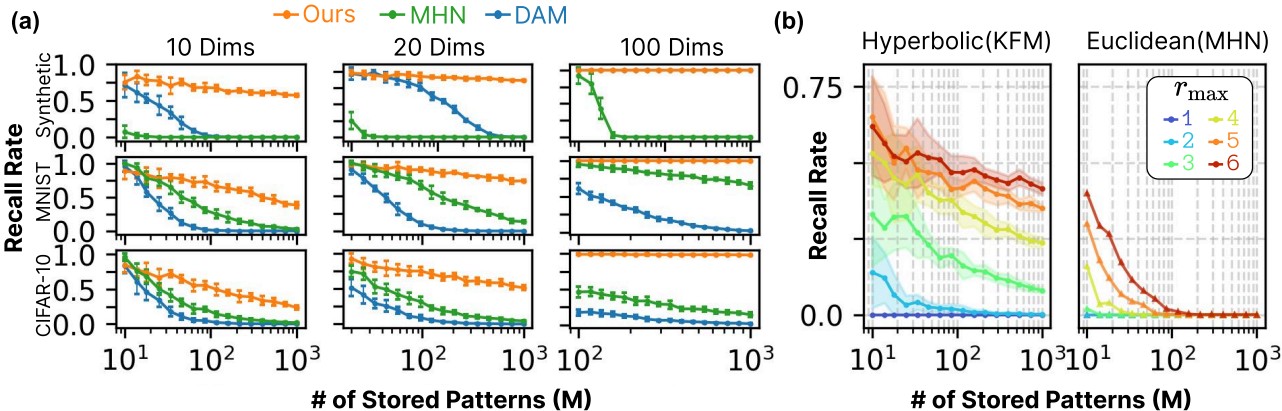

*Figure 2.* **(a) Left to right columns:** pattern dimension $d \in \{10, 20, 100\}$. **(a) Top to bottom rows:** Recall success rate of three models on *synthetic, MNIST, CIFAR10* datasets. We observe that the Karcher-flow model outperforms other models on the synthetic, MNIST, and CIFAR10 datasets with superior scaling. **(b): Left to right:** The recall rate of the Karcher-flow model and the MHN under different values of $r_{\max}$, when $d = 3$, respectively. The stored patterns are sampled from a ball of radius $r_{\max}$. MHN shows marginal improvement on recall rate while our model benefits from larger $r_{\max}$.

Hopfield, 2021) and Modern Hopfield Networks (MHN) (Ramsauer et al., 2020), two well-studied memory models for the continuous domain. For ML tasks, we compare against the Hopfield-based layers proposed in (Ramsauer et al., 2020). Source code is available on GitHub.

## 5.1. Pattern Completion

The task involves recalling a memory pattern from a stored memory set. We aim to reconstruct memories based on a query that is a corrupted/noisy version of the target memory. We assume that all memory patterns lie in $T_o \mathbb{H}$. We also conduct analysis where we fix $d = 3$, where the state pattern is represented by only 3 neurons. We observe how different values of $r_{\max}$ affect memory capacity.

**Setup and Evaluation Metrics.** For memory patterns, we utilize: (1) synthetic points uniformly sampled within a tangent ball, (2) MNIST (LeCun et al., 2002), and (3) CIFAR10 (Krizhevsky et al., 2009). For the Karcher-flow model, we follow the query generation process described in Section 4 to generate the corresponding queries in $\mathbb{H}$. For each trial, we sweep through different values of $M$ (the memory set size) and perform pattern completion, treating each pattern in the set as a target in turn. A successful recall means the retrieval error is less than a tolerance 0.01 under either hyperbolic or Euclidean metric. The recall success rate is then calculated for each value of $M$.

**Results.** The simulation results are presented in Figure 2. In Figure 2(a), we report results when keeping $d \in \{10, 20, 100\}$ PCA dimensions with $r_{\max} = 3$. We repeated the task across 10 different random seeds and report the mean and standard deviation. For Figure 2(b), we vary $r_{\max}$ from 1 to 6. We observe that our Karcher-flow model demonstrates a high recall success rate, whereas the

other two baselines fail to store even a small number of patterns in low-dimensional settings. As shown in the left panel of Figure 2(b), our model exhibits significant capacity improvement as $r_{\max}$ increases. In contrast, the results in the right panel of Figure 2(b) show that MHN do not benefit much from this rescaling operation. Additional results are provided in Section G.1.

## 5.2. Classification

Inspired by Ramsauer et al. (2020), who parameterized MHNs and proposed various machine learning layers, we describe our proposed layers in Section F.3. Interestingly, we can construct these layers without any hyperbolic parameters. This allows our model to benefit from superior capacity scaling while still utilizing Euclidean optimizers. We also include two existing hyperbolic attention modules: (1) hyperbolic attention networks (Gulcehre et al., 2019), and (2) hyperbolic neural networks ++ (Shimizu et al., 2021). A major difference between (Shimizu et al., 2021) and other models is that it requires a Riemannian-based optimizer as the parameters are defined in the hyperbolic space.

**Setup and Evaluation Metrics.** We evaluate the layers across both image classification and multiple instance learning (MIL) benchmarks. For image classification, we use MNIST (LeCun et al., 2002). For MIL we use the Elephant, Fox, and Tiger datasets. We construct our network with one embedding layer, followed by the layer of interest, and a readout layer. The pooling layers have a single learnable static query. We use the AdamW optimizer (Loshchilov & Hutter, 2019) for training, and hyperparameters are provided in Table 6. We report mean accuracy with standard deviation over 5 trials for classification, and mean ROC AUC across trials under 10-fold cross-validation for MIL.

*Table 1.* Performance comparison across MNIST and MIL benchmarks. MNIST results use `KFAttention` and `Hopfield` attention mechanisms evaluated across hidden dimensions $d$, while MIL results use the corresponding pooling variants (`KFPooling` and `HopfieldPooling`) on three animal datasets. Values are reported as mean $\pm$ standard deviation.

| Model | MNIST (% accuracy) | | | MIL (AUC) | | |
| --- | --- | --- | --- | --- | --- | --- |
| | $d = 4$ | $d = 8$ | $d = 32$ | Tiger | Fox | Elephant |
| `KarcherFlow` | $\mathbf{85.52 \pm 2.50}$ | $\mathbf{92.42 \pm 0.64}$ | $\mathbf{96.89 \pm 0.13}$ | $87.34 \pm 1.40$ | $\mathbf{66.00 \pm 2.32}$ | $91.20 \pm 0.82$ |
| `Hopfield` | $83.70 \pm 2.12$ | $92.29 \pm 0.63$ | $96.71 \pm 0.21$ | $83.52 \pm 1.50$ | $60.54 \pm 2.58$ | $91.65 \pm 0.42$ |
| (Gulcehre et al., 2019) | $84.85 \pm 1.85$ | $91.71 \pm 1.07$ | $96.80 \pm 0.18$ | $\mathbf{89.20 \pm 0.94}$ | $62.92 \pm 2.41$ | $\mathbf{93.04 \pm 0.42}$ |
| (Shimizu et al., 2021) | $67.35 \pm 5.76$ | $84.17 \pm 7.21$ | $84.17 \pm 6.12$ | $80.32 \pm 5.11$ | $57.76 \pm 4.84$ | $85.32 \pm 3.43$ |

**Results.** On MNIST, Karcher flow–based models achieve performance comparable to Hopfield models under high hidden dimensions. However, Karcher flow models consistently yield higher accuracy in low-dimensional representation spaces, as shown in Table 1. Similar trends have been reported in other implementations of hyperbolic attention networks when compared against their Euclidean counterparts (Gulcehre et al., 2019; Zhang et al., 2022). On the Elephant, Fox, and Tiger MIL benchmarks, `KFPooling` or previous hyperbolic attention networks consistently outperformed Euclidean `HopfieldPooling` models in terms of ROC AUC, but other hyperbolic networks sometimes outperformed `KFPooling`. See MIL results in Table 1.

## 6. Discussion

Here we discuss three questions: (i) how hyperbolic geometry arises from neural population activity, (ii) what this geometry implies for downstream decoding, and (iii) how to transfer this insight to machine learning. Our theoretical results and the simulations of the previous sections answer these in turn, and we now discuss what each one says.

**Hyperbolic Structure in Neural Population Codes.** First, our definition of $d_{ab} = -\phi(\langle \lambda(s_a), \lambda(s_b) \rangle) + C$ follows the standard RSA convention in computational neuroscience (Kriegeskorte et al., 2008). We theoretically show that the semi-metric between stimuli is statistically hyperbolic. In other words, with the number of active neurons scales linearly with the size of the environment, the semi-metric is 0-hyperbolic, which corresponds to a tree metric. This result is informative for two reasons. (1) By Kriegeskorte & Wei (2021), tuning curves define the geometry, which informs how the downstream decoding process can exploit hierarchical relationships. (2) Furthermore, this result provides a theoretical foundation for the findings in (Zhang et al., 2022), suggesting that the cognitive map may have a tree-like organization.

**A Hyperbolic Associative Memory Model.** In Proposition 2.2, we link memory recall and decoding by showing that the Modern Hopfield Network (MHN) computes the optimal Minimum Mean Square Error (MMSE) estimator for decoding. Combining this connection with Theorem 4.2

allows us to study the corresponding decoder, viewed as an associative memory model under hyperbolic geometry. Theoretically, Theorem 4.8 shows that our Karcher-flow model exhibits an extra *double-exponential scaling* term in $r_{\max}$ (the maximum norm among memory patterns). This scaling behavior is not present in existing Euclidean-based models (Ramsauer et al., 2020; Krotov & Hopfield, 2021; Krotov, 2023). Specifically, the double-exponential term arises for two reasons: (1) hyperbolic geometry exhibits exponential volume growth with respect to its radius, and (2) the Lorentz inner product naturally encodes a term approximately proportional to $e^{d_g(a,b)}$. Interestingly, the Lorentz inner product is a linear operation and requires the same computational complexity as the Euclidean inner product.

**Simulation Results.** We validate Theorem 4.8 via pattern completion in Figure 2. For our case study in Figure 2(b), rescaling the memory patterns corresponds to rescaling the synaptic connection strength between neurons. Figure 2(b) shows that both DAM and MHN, do not benefit much from this modification. The traditional approach in AM to increase model capacity is to increase network width or depth. Notably, our model offers an additional architectural perspective, obtaining performance gains without any modification to width or depth. Furthermore, our model offers an additional perspective on how systems with a small number of neurons can store many states.

**Future Directions.** This work suggests several avenues for future research. In computational neuroscience, an interesting direction is to understand how synaptic plasticity rules in the hippocampus either promote or constrain the emergence of this geometry. Additionally, our theory could be generalized to the multi-field case to determine how complex tuning curves induce hyperbolic structure. From a machine learning perspective, our proof-of-concept simulations demonstrate that these bio-inspired layers outperform their Euclidean counterparts in classification and Multiple Instance Learning (MIL) tasks, particularly in low-dimensional regimes. A promising future direction involves scaling these hyperbolic architectures to larger scales, and discovering suitable application scenarios.

## Acknowledgements

Dennis Wu is supported by the Northwestern Cognitive Science Graduate Fellowships for Interdisciplinary Research Projects. Yi-Chun Hung is supported by the Taiwan-Northwestern Doctoral Scholarship. James Fitzgerald is supported by the National Institute for Theory and Mathematics in Biology through the National Science Foundation (grant number DMS-2235451) and the Simons Foundation (grant number MPTMPS-00005320). Han Liu is partially supported by NIH R01LM1372201, NSF AST-2421845, Simons Foundation MPS-AI-00010513, AbbVie, Dolby and Chan Zuckerberg Biohub Chicago Spoke Award. This research was supported in part through the computational resources and staff contributions provided for the Quest high performance computing facility at Northwestern University which is jointly supported by the Office of the Provost, the Office for Research, and Northwestern University Information Technology.

## Impact Statement

From the perspective of (Bender & Hanna, 2025), the reliance on massive computational resources concentrates power within a few large companies (e.g., Google). Our method introduces an additional dimension—hyperbolic geometry—while keeping the overall model size unchanged. Despite this added complexity, the required computational resources remain comparable, yet the model demonstrates superior memory capacity. This improvement in efficiency could lower the barrier to entry, enabling smaller companies and research labs to develop competitive models, thereby helping to reduce power concentration in AI research.

From the perspective of (Crawford & Calo, 2016), current LLMs largely depend on scaling laws to increase intelligence, which leads to significant carbon emissions. While our model does not fully resolve this issue, it provides a promising, under-explored approach for energy-efficient AI. As shown in Theorem 4.8, our model achieves double-exponential scaling in memory capacity, allowing efficient performance even in strictly low-dimensional settings. This offers a theoretical foundation for designing more energy-conscious architectures. However, we acknowledge potential negative consequences. Efficiency gains in memory and representation may enable deployment of ML models in resource-constrained devices, which could facilitate surveillance or autonomous weapons systems. Furthermore, consistent with Jevons Paradox, improvements in computational efficiency may paradoxically increase overall energy consumption by lowering the barrier to large-scale deployment.

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

# Supplementary Materials

**LLM Usage Disclosure.** We use AI tools such as LLMs to aid with polishing writing including improving clarity and grammar. All the technical results are original contributions by the authors.

## A. Limitations

Our tuning-curve model assumes single-field neurons and uniform peak firing rates. However, these properties can be violated in large environments (Rich et al., 2014). In the neuroscience setting, it would also be interesting to go beyond the hyperboloid model to calculate memory capacities when the embedding dimension far exceeds the dimensionality of the hyperbolic manifold. It is unclear whether the hyperbolic structure we identify is a consequence of more general constraints in biology or a phenomenon specific to hippocampal place cells. Distinguishing these would clarify whether our framework generalizes to other brain regions. For instance, hyperbolic geometry has also been reported in the olfactory and visual systems (Zhou et al., 2018; Lee et al., 2024).

# B. Table of Notations

To distinguish between hyperbolic and Euclidean points, we use regular font for Euclidean variables and bold font for hyperbolic variables.

*Table 2.* Mathematical Notations and Symbols

| Symbol | Description |
|---|---|
| *Indices and dimensions* | |
| $N$ | Number of neurons (index $i \in [N]$; Section 2) |
| $M$ | Number of stored patterns or grid points (index $\mu \in [M]$) |
| $d$ | Latent hyperbolic embedding dimension ($\mathbb{H}_\kappa^d$) |
| $D$ | Dimension of stimulus domain $\mathcal{S} \subset \mathbb{R}^D$ |
| *Stimulus space, spikes, and tuning* | |
| $\mathcal{S}$ | Stimulus space (e.g. $\mathcal{S} = [0, L]^D$ in Theorem 4.2) |
| $s$ | Stimulus (location) |
| $n = (n_1, \ldots, n_N)$ | Spike counts in a window (Section 2) |
| $\lambda_i(s)$ | Tuning curve of neuron $i$ (Equation (2.1)) |
| $\lambda(s)$ | Population rate vector $(\lambda_1(s), \ldots, \lambda_N(s))$ |
| $s_{ik}, \ \sigma_{ik}, \ \lambda_{ik}$ | Center, width, and scale of the $k$-th field of neuron $i$ (Equation (2.1)) |
| $K$ | Number of place fields per neuron (often $K = 1$; Equation (2.1)) |
| $\lambda_{\max}$ | Peak firing rate (uniform-field simplification; Equation (2.1)) |
| $\rho$ | Intensity of Poisson field centers (Theorem 4.2) |
| *Riemannian hyperboloid model* | |
| $(\mathcal{M}, g)$ | Riemannian manifold with metric $g$ (Hadamard in our setting) |
| $T_x \mathcal{M}$ | Tangent space at $x \in \mathcal{M}$ |
| $d_g(x, y)$ | Geodesic distance between $x, y \in \mathcal{M}$ |
| $\mathrm{Exp}_x(\cdot), \ \mathrm{Exp}_x^{-1}(\cdot)$ | Exponential map and inverse at $x$ |
| $\mathbb{H}_\kappa^d$ | Hyperboloid (Lorentz) model, curvature $\kappa < 0$ |
| $\langle \cdot, \cdot \rangle_L$ | Lorentz (Minkowski) inner product |
| $\boldsymbol{o}$ | Origin (reference point) on $\mathbb{H}_\kappa^d$ |
| $\boldsymbol{v}$ | Query / state on $\mathbb{H}_\kappa^d$ |
| $\mathrm{v}$ | Log-coordinates at $\boldsymbol{o}$: $\mathrm{v} = \mathrm{Exp}_{\boldsymbol{o}}^{-1}(\boldsymbol{v})$ |
| $\boldsymbol{\sigma}$ | Noise variance applied on $\boldsymbol{v}$ |
| *Memory patterns and maps* | |
| $\psi^H : \mathbb{R}^N \to \mathbb{H}_\kappa^d$ | Feature map from firing rate to hyperbolic space (Section 4) |
| $\psi^E : \mathbb{R}^N \to R^d$ | Feature map from firing rate to Euclidean space (Section 4) |
| $\xi_\mu$ | $\mu$-th stored memory pattern in $\mathbb{R}^d$ |
| $\boldsymbol{\xi}_\mu$ | $\mu$-th stored memory pattern in $\mathbb{H}_\kappa^d$ |
| $p(\mu \mid n)$ | Posterior over memory index given spikes (Section 4) |
| $w_\mu(\boldsymbol{v})$ | Softmax weights in the Karcher-flow update (KFM) |
| $H(\boldsymbol{v})$ | Karcher-flow update rule (KFM) |
| *Hyperbolicity* | |
| $d_{ab}$ | Semi-metric on stimuli from population inner products (Section 4) |
| $\Delta(s_a, s_b, s_c, s_d)$ | Empirical Four-point hyperbolicity (Theorem 4.2) |
| $\delta$ | Hyperbolicity scale (Theorem 3.2) |
| $L$ | Stimulus space diameter $\mathcal{S} = [0, L]^D$ (Theorem 4.2) |

## C. Related Works

**Associative Memory Models.** The foundations of associative memory models were established in the works of (Amari, 1972; Nakano, 2007; Amari & Maginu, 1988; Hopfield, 1982). The classic Hopfield network (Hopfield, 1982) is one of the most well-studied associative memory models due to its energy-based structure and theoretical accessibility. Recent studies (Krotov & Hopfield, 2021; Ramsauer et al., 2020) substantially increased the storage capacity of Hopfield networks and generalized associative memory models to continuous domains. More recently, modern Hopfield networks have been shown to be closely related to attention mechanisms in transformers (Ramsauer et al., 2020; Vaswani et al., 2017). Variants of modern Hopfield networks were then proposed to both achieve higher capacity and recover different types of attention mechanisms (Hu et al., 2023; Santos et al., 2025; Burns & Fukai, 2023; Krotov, 2023; Wu et al., 2024).

**Neural Population Geometry.** In computational neuroscience, the geometric aspects of neural population activities have gained growing attention in recent years (Kriegeskorte & Wei, 2021). Most works study the activity patterns of animals when they encounter different stimuli. For example, Chapman & Störmer (2024) studies population geometry of the visual system to analyze which visual features animals attend to. Panzeri et al. (2022) studies how neural populations represent and process information by analyzing the activity correlations between place cells in the hippocampus. Besides studying the structure of neural representation spaces, researchers also try to understand the intrinsic geometry in those spaces. Specifically, growing evidence shows that hyperbolic geometry naturally occurs in various biological systems (Kagel & Sharpee, 2025; Lecca et al., 2023; Sharpee, 2019; Lee et al., 2024; Allard & Serrano, 2020). Zhang et al. (2023) shows that the semi-metric between hippocampal place cells has the same topological structure as hyperbolic geometry, suggesting a hyperbolic cognitive map in animal brains for spatial information. Next, they show that when animals gain experience, the radius of the latent cognitive map grows larger. This finding provides a unique angle to connect learning to/with neural population geometry. Similarly, Zhou et al. (2018); Ghaninia et al. (2022) find that besides spatial information, animals also encode odors into a latent hyperbolic map using a similar analytical tool used in (Zhang et al., 2023). There are two major differences between our work and (Zhang et al., 2023; Zhou et al., 2018; Ghaninia et al., 2022). First, our results are mainly theoretical, whereas previous works obtain their results empirically using topological data analysis. Next, their studies focused on showing that neuron-to-neuron correlations are topologically hyperbolic, hereas we instead show that inter-stimuli distances are hyperbolic from a metric perspective.

**Methods for Geometric Analysis of Neural Representations.** Various algorithms have been proposed to (1) measure the underlying geometry of neural representations and (2) compare the similarities between two sets of neural responses. For (1), a typical example is Zhang et al. (2023), where they utilize a technique in topological data analysis (TDA) called the Betti curve. In topology, the Betti numbers are used to distinguish topological spaces based on the number of $n$-dimensional holes in the space. In Zhang et al. (2023), the authors compute the pair-wise dissimilarity between neurons, and construct a graph with vertices being the neurons. The vertices are connected if their pair-wise dissimilarity is below some threshold $\tau$. The Betti number is computed for each $\tau$, which forms the Betti curve for each Betti number. This method allows researchers to compare experimental data to well-defined topological spaces.

# D. Additional Background

## D.1. Visualization of $\delta$-thin Triangles

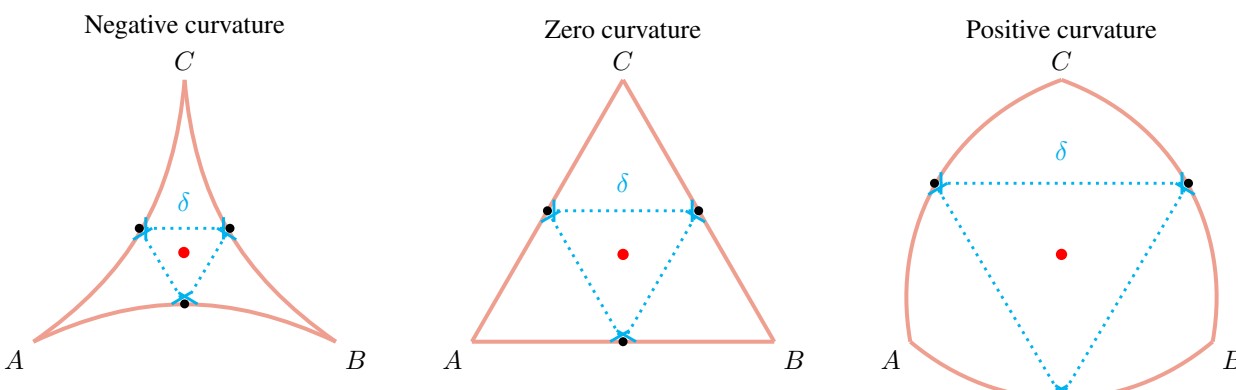

*Figure 3. $\delta$-thin triangles:* the $\delta$-thin triangles refer to the geodesic triangles formed by points $A, B, C$ connected with orange geodesics. For each triangle, the red dot denotes its center, black dots refer to the midpoints of its sides, cyan dot lines measures the length between any two midpoints. Hyperbolic spaces exhibit uniformly thin triangles (small $\delta$), Euclidean spaces are borderline, and spherical/positively curved spaces exhibit "fatter" triangles (larger $\delta$). The arrows indicate representative distances from midpoints of each side to the center of the circle. The diameter of this set of midpoints is bounded by $\delta$, and the circle's radius gives $\frac{\delta}{2}$. on one side to the union of the other two sides, all of which are uniformly bounded by $\delta$. The figure is schematic and not drawn to scale.

## D.2. Properties of Hyperbolic Geometry

**Lemma D.1.** *Let $\kappa < 0$, $\alpha := \sqrt{|\kappa|}$, and let*

$$\boldsymbol{x} = \mathrm{Exp}_o(a), \qquad \boldsymbol{y} = \mathrm{Exp}_o(b), \qquad \boldsymbol{z} = \mathrm{Exp}_o(c)$$

*be hyperboloid embeddings of $a, b, c \in T_o\mathcal{M} \simeq \mathbb{R}^d$. Define the squared Euclidean distance gap*

$$\Delta^{\mathrm{E}}(a; b, c) := \|a - c\|_2^2 - \|a - b\|_2^2,$$

*and let*

$$g(r) := \frac{\sinh(\alpha r)}{r}, \qquad r > 0,$$

*with $g(0) = \alpha$.*

*Then the Lorentz inner-product difference admits the exact decomposition*

$$\langle \boldsymbol{x}, \boldsymbol{y} \rangle_L - \langle \boldsymbol{x}, \boldsymbol{z} \rangle_L = \frac{1}{|\kappa|} g(\|a\|_2) \frac{g(\|b\|_2)}{2} \Delta^{\mathrm{E}}(a; b, c) - \mathrm{Pen}_\kappa(a; b, c), \tag{D.1}$$

*where the penalty term $\mathrm{Pen}_\kappa(a; b, c)$ is given by*

$$\mathrm{Pen}_\kappa(a; b, c) := \frac{1}{|\kappa|} \cosh(\alpha\|a\|_2) \Big( \cosh(\alpha\|b\|_2) - \cosh(\alpha\|c\|_2) \Big) \tag{D.2}$$

$$+ \frac{1}{|\kappa|} g(\|a\|_2) \left[ \frac{g(\|b\|_2)}{2} \big( \|c\|_2^2 - \|b\|_2^2 \big) - \big( g(\|b\|_2) - g(\|c\|_2) \big) \langle a, c \rangle \right]. \tag{D.3}$$

*Moreover, if $\|a\|_2, \|b\|_2, \|c\|_2 \in [r_{\min}, r_{\max}]$ with $0 < r_{\min} \leq r_{\max} < \infty$, then*

$$\langle \boldsymbol{x}, \boldsymbol{y} \rangle_L - \langle \boldsymbol{x}, \boldsymbol{z} \rangle_L \geq A_\kappa(r_{\min}; a) \Delta^{\mathrm{E}}(a; b, c) - \mathrm{Pen}_\kappa(r_{\min}, r_{\max}), \tag{D.4}$$

*where*

$$A_\kappa(r_{\min}; a) := \frac{1}{|\kappa|} g(\|a\|_2) \frac{g(r_{\min})}{2},$$

*and the uniform penalty bound*

$$\mathsf{Pen}_\kappa(r_{\min}, r_{\max}) \;\coloneqq\; \sup_{\|a\|_2, \|b\|_2, \|c\|_2 \in [r_{\min}, r_{\max}]} \mathsf{Pen}_\kappa(a; b, c)$$

*is finite and satisfies the rate*

$$\mathsf{Pen}_\kappa(r_{\min}, r_{\max}) = \mathcal{O}\left(\frac{1}{|\kappa|} e^{2\alpha r_{\max}} r_{\max}^2\right), \qquad \alpha = \sqrt{|\kappa|},$$

*with constants depending only on* $r_{\min} > 0$.

*Proof of Lemma D.1.* We rewrite the Lorentz inner-product difference in terms of the *squared Euclidean distance gap*

$$\Delta^{\mathrm{E}}(a; b, c) \;\coloneqq\; \|a - c\|_2^2 - \|a - b\|_2^2.$$

Let $\alpha \coloneqq \sqrt{|\kappa|}$ and $s^2 = 1/|\kappa|$. Under the hyperboloid model, for any $u \in T_o\mathcal{M} \simeq \mathbb{R}^d$,

$$\mathrm{Exp}_o(u) = \left(s \cosh(\alpha\|u\|_2), \; s\,\frac{\sinh(\alpha\|u\|_2)}{\|u\|_2}\, u\right).$$

Define

$$\boldsymbol{x} \coloneqq \mathrm{Exp}_o(a), \qquad \boldsymbol{y} \coloneqq \mathrm{Exp}_o(b), \qquad \boldsymbol{z} \coloneqq \mathrm{Exp}_o(c),$$

and for convenience set

$$g(r) \coloneqq \frac{\sinh(\alpha r)}{r}, \qquad r > 0,$$

(with $g(0) = \alpha$). A direct substitution gives the exact identity

$$\langle \boldsymbol{x}, \boldsymbol{y} \rangle_L - \langle \boldsymbol{x}, \boldsymbol{z} \rangle_L = -\frac{1}{|\kappa|} \cosh(\alpha\|a\|_2)\Big(\cosh(\alpha\|b\|_2) - \cosh(\alpha\|c\|_2)\Big) \tag{D.5}$$

$$+ \frac{1}{|\kappa|} g(\|a\|_2)\Big(g(\|b\|_2)\,\langle a, b\rangle - g(\|c\|_2)\,\langle a, c\rangle\Big). \tag{D.6}$$

By expanding squared norms,

$$\Delta^{\mathrm{E}}(a; b, c) = \big(\|a\|_2^2 + \|c\|_2^2 - 2\langle a, c\rangle\big) - \big(\|a\|_2^2 + \|b\|_2^2 - 2\langle a, b\rangle\big)$$
$$= 2\big(\langle a, b\rangle - \langle a, c\rangle\big) + \big(\|c\|_2^2 - \|b\|_2^2\big).$$

Hence

$$\langle a, b\rangle - \langle a, c\rangle = \frac{1}{2}\Delta^{\mathrm{E}}(a; b, c) - \frac{1}{2}\big(\|c\|_2^2 - \|b\|_2^2\big). \tag{D.7}$$

Rewrite the bracket in (D.6) as

$$g(\|b\|_2)\,\langle a, b\rangle - g(\|c\|_2)\,\langle a, c\rangle = g(\|b\|_2)\big(\langle a, b\rangle - \langle a, c\rangle\big) + \big(g(\|b\|_2) - g(\|c\|_2)\big)\langle a, c\rangle.$$

Substituting (D.7) yields

$$g(\|b\|_2)\,\langle a, b\rangle - g(\|c\|_2)\,\langle a, c\rangle = \frac{g(\|b\|_2)}{2}\Delta^{\mathrm{E}}(a; b, c) - \frac{g(\|b\|_2)}{2}\big(\|c\|_2^2 - \|b\|_2^2\big) + \big(g(\|b\|_2) - g(\|c\|_2)\big)\langle a, c\rangle. \tag{D.8}$$

Plugging (D.8) into (D.6) and combining with (D.5) gives the *exact decomposition*

$$\langle \boldsymbol{x}, \boldsymbol{y} \rangle_L - \langle \boldsymbol{x}, \boldsymbol{z} \rangle_L = \underbrace{\frac{1}{|\kappa|} g(\|a\|_2)\,\frac{g(\|b\|_2)}{2}\,\Delta^{\mathrm{E}}(a; b, c)}_{\text{main term}} - \underbrace{\mathsf{Pen}_\kappa(a; b, c)}_{\text{penalty}}, \tag{D.9}$$

where the penalty $\mathsf{Pen}_\kappa(a; b, c)$ is given explicitly by

$$\mathsf{Pen}_\kappa(a; b, c) := \frac{1}{|\kappa|} \cosh(\alpha\|a\|_2) \Big( \cosh(\alpha\|b\|_2) - \cosh(\alpha\|c\|_2) \Big) \tag{D.10}$$

$$+ \frac{1}{|\kappa|} g(\|a\|_2) \left[ \frac{g(\|b\|_2)}{2} \big( \|c\|_2^2 - \|b\|_2^2 \big) - \big( g(\|b\|_2) - g(\|c\|_2) \big) \langle a, c \rangle \right]. \tag{D.11}$$

Equations (D.9)–(D.11) are obtained from the closed-form hyperboloid exponential map.

Assume $\|a\|_2, \|b\|_2, \|c\|_2 \in [r_{\min}, r_{\max}]$ with $0 < r_{\min} \le r_{\max} < \infty$. Since $g(\cdot)$ is increasing on $(0, \infty)$, we have $g(\|b\|_2) \ge g(r_{\min})$, and thus the main term in (D.9) admits the uniform lower bound

$$\frac{1}{|\kappa|} g(\|a\|_2) \frac{g(\|b\|_2)}{2} \Delta^{\mathrm{E}}(a; b, c) \ge \frac{1}{|\kappa|} g(\|a\|_2) \frac{g(r_{\min})}{2} \Delta^{\mathrm{E}}(a; b, c).$$

Define the amplification factor

$$A_\kappa(r_{\min}; a) := \frac{1}{|\kappa|} g(\|a\|_2) \frac{g(r_{\min})}{2}.$$

Then (D.9) implies the deterministic inequality

$$\langle \boldsymbol{x}, \boldsymbol{y} \rangle_L - \langle \boldsymbol{x}, \boldsymbol{z} \rangle_L \ge A_\kappa(r_{\min}; a) \Delta^{\mathrm{E}}(a; b, c) - \sup_{\|a\|_2, \|b\|_2, \|c\|_2 \in [r_{\min}, r_{\max}]} \mathsf{Pen}_\kappa(a; b, c). \tag{D.12}$$

Finally, note that the supremum penalty term in (D.12) is finite for fixed $(r_{\min}, r_{\max}, \kappa)$ and can be bounded explicitly using $\cosh(\alpha r) \lesssim e^{\alpha r}$ and $g(r) = \sinh(\alpha r)/r \lesssim e^{\alpha r}/r$ on $[r_{\min}, r_{\max}]$. In particular, one obtains the coarse rate

$$\sup \mathsf{Pen}_\kappa(a; b, c) = \mathcal{O}\left( \frac{1}{|\kappa|} e^{2\alpha r_{\max}} r_{\max}^2 \right), \qquad \alpha = \sqrt{|\kappa|},$$

where the hidden constant depends only on $r_{\min} > 0$ (through $1/r_{\min}$ factors in $g$ and its Lipschitz constant on the interval). $\qquad\square$

**Lemma D.2.** *Let $\kappa < 0$, $\alpha = \sqrt{|\kappa|}$, and $\boldsymbol{v} = \mathrm{Exp}_{\boldsymbol{o}}(\mathrm{v})$ with $\mathrm{v} \in \mathbb{R}^d$. Let $\boldsymbol{\xi}_\rho = \mathrm{Exp}_{\boldsymbol{o}}(s_\rho)$ for all $\rho \in [M]$. Fix $\mu \ne \nu$ and define the squared-distance gap*

$$\Delta^{\mathrm{E}}_{\mu\nu}(\mathrm{v}) := \|\mathrm{v} - s_\nu\|_2^2 - \|\mathrm{v} - s_\mu\|_2^2.$$

*Assume $\|\mathrm{v}\|_2, \|s_\mu\|_2, \|s_\nu\|_2 \in [r_{\min}, r_{\max}]$. Then*

$$\Delta^{\mathrm{H}}_{\mu\nu}(\boldsymbol{v}) = \langle \boldsymbol{v}, \boldsymbol{\xi}_\mu \rangle_L - \langle \boldsymbol{v}, \boldsymbol{\xi}_\nu \rangle_L \ge f_\kappa(r_{\min}) \Delta^{\mathrm{E}}_{\mu\nu}(\mathrm{v}) - \tilde{\mathsf{Pen}}_\kappa(r_{\min}, r_{\max}),$$

*where*

$$f_\kappa(r) := \frac{1}{2|\kappa|} \left( \frac{\sinh(\alpha r)}{r} \right)^2,$$

*and $\tilde{\mathsf{Pen}}_\kappa(r_{\min}, r_{\max})$ is the uniform penalty bound from Corollary Theorem D.3.*

*Proof.* Apply Lemma D.1 with $a = \mathrm{v}$, $b = s_\mu$, $c = s_\nu$ and $(x, y, z) = (\boldsymbol{v}, \boldsymbol{\xi}_\mu, \boldsymbol{\xi}_\nu)$. Then use the uniformization on $[r_{\min}, r_{\max}]$ and the definition of $f_\kappa$. $\qquad\square$

**Corollary D.3.** *Let $\kappa < 0$, $\alpha := \sqrt{|\kappa|}$, and assume the stored patterns satisfy $r_{\min} \le \|s_\mu\|_2 \le r_{\max}$ for all $\mu$, with $\Delta r := r_{\max} - r_{\min}$. Define*

$$\Delta^{\mathrm{E}}_\mu := \min_{\nu \ne \mu} \|s_\mu - s_\nu\|_2^2, \qquad \Delta^{\mathrm{H}}_\mu := \min_{\nu \ne \mu} \Big( \langle \xi_\mu, \xi_\mu \rangle_L - \langle \xi_\mu, \xi_\nu \rangle_L \Big).$$

*Then for each $\mu$,*

$$\Delta^{\mathrm{H}}_\mu \ge f_\kappa(r_{\min}) \Delta^{\mathrm{E}}_\mu - \tilde{\mathsf{Pen}}_\kappa(r_{\min}, r_{\max}),$$

*where $f_\kappa(r) = \frac{1}{2|\kappa|}\left(\frac{\sinh(\alpha r)}{r}\right)^2$, and the penalty admits the bound*

$$\widetilde{\mathrm{Pen}}_\kappa(r_{\min}, r_{\max}) \;\leq\; \frac{1}{|\kappa|}\left[\alpha \cosh^2(\alpha r_{\max})\,\Delta r \;+\; C_g(r_{\min}, r_{\max})\,g(r_{\max})\,r_{\max}^2\,\Delta r\right],$$

*with $g(r) = \sinh(\alpha r)/r$ and*

$$C_g(r_{\min}, r_{\max}) \;:=\; \sup_{r \in [r_{\min}, r_{\max}]} \left|g'(r)\right|.$$

*In particular, since $\cosh(\alpha r) \lesssim e^{\alpha r}$ and $g(r) \lesssim e^{\alpha r}/r$ on $[r_{\min}, r_{\max}]$, we also have the following rate*

$$\widetilde{\mathrm{Pen}}_\kappa(r_{\min}, r_{\max}) = \mathcal{O}\left(\frac{1}{|\kappa|}\,e^{2\alpha r_{\max}}\,r_{\max}^2\,\Delta r\right), \qquad \Delta r = r_{\max} - r_{\min}.$$

## D.3. Technical Tools

**Definition D.4.** For a point process $\boldsymbol{X}$, its intensity measure $\nu$, evaluated on a Borel set $B$, is defined as

$$\nu(B) = \mathbb{E}[\boldsymbol{X}(B)].$$

**Theorem D.5** (Campbell's Theorem (Baddeley et al., 2007)). *Let $\boldsymbol{X}$ be a point process on $S$ and let $f : S \to \mathbb{R}$ be a measurable function. Then the random sum*

$$T \coloneqq \sum_{x \in \boldsymbol{X}} f(x)$$

*is a random variable, with expected value*

$$\mathbb{E}\left[\sum_{x \in \boldsymbol{X}} f(x)\right] = \int_S f(x)\,\nu(dx).$$

*Further, if $\boldsymbol{X}$ is a point process on $\mathbb{R}^d$ with constant intensity $\rho$, the expectation becomes*

$$\mathbb{E}\left[\sum_{x \in \boldsymbol{X}} f(x)\right] = \rho \int_{\mathbb{R}^d} f(x)\,dx,$$

where $\nu(dx) = \rho dx$.

**Lemma D.6** (Markov's Inequality). *Let $X$ be a nonnegative random variable and $a > 0$. Then*

$$\Pr[X \geq a] \leq \frac{\mathbb{E}[X]}{a}.$$

# E. Proof of Main Text Results

## E.1. Proof of Proposition 2.2

*Proof.* The objective is convex in $z$. Differentiating and setting to zero:

$$\frac{d}{dz}\,\mathbb{E}_{p(\mu|v)}\left[\|\xi_\mu - z\|_2^2\right] = -2\sum_{\mu=1}^{M} p(\mu \mid v)\,(\xi_\mu - z) = 0. \tag{E.1}$$

Solving directly:

$$z^* = \sum_{\mu=1}^{M} p(\mu \mid v)\,\xi_\mu. \tag{E.2}$$

Substituting the Boltzmann posterior:

$$z^* = \sum_{\mu=1}^{M} \frac{e^{\langle v, \xi_\mu \rangle}}{\sum_\nu e^{\langle v, \xi_\nu \rangle}}\,\xi_\mu = \sum_{\mu=1}^{M} \mathrm{softmax}_\mu\left(h_1^{\mathrm{MHN}}(v), \ldots, h_M^{\mathrm{MHN}}(v)\right)\xi_\mu = \mathrm{MHN}(v). \tag{E.3}$$

$\square$

## E.2. Proof of Theorem 4.2

Define the kernel

$$K(s, s') := \sum_{i=1}^{N} \lambda_i(s)\lambda_i(s'), \quad \lambda_i(s) = \exp\left(-\frac{||s - s_i||_2^2}{2\sigma_i^2}\right).$$

Define the corresponding metric-like distance

$$d(s, s') = -\ln\left(K(s, s')\right).$$

Our goal is to show that for any 4 points $(s_a, s_b, s_c, s_d)$, the quantity

$$\Delta = d(s_a, s_b) + d(s_c, s_d) - \max\{d(s_a, s_c) + d(s_b, s_d), d(s_a, s_d) + d(s_b, s_c)\},$$

is bounded by a constant $2\delta$ with high probability.

**Definition E.1.**

$$K_{\max}(s, s') := \max_i \langle \lambda_i(s), \lambda_i(s') \rangle, \quad d^*(s, s') := -\ln(K_{\max}(s, s')).$$

**Definition E.2.** Let $i := \arg\max_{j \in [N]} \sigma_j$, then

$$d_i(s, s') := -\ln(\lambda_i(s)\lambda_i(s')).$$

**Lemma E.3.** *For any $j \in [N]$, $d_j$ is 0-hyperbolic.*

*Proof.* Observe

$$\begin{aligned}
d_j(s, s') &= -\ln\left(\lambda_j(s)\lambda_j(s')\right) \\
&= \frac{1}{2\sigma_j^2}(||s - s_j||_2^2 + ||s' - s_j||_2^2) \\
&= a_j(w_j(s) + w_j(s')),
\end{aligned}$$

where $a_j = \frac{1}{2\sigma_j^2}$, and $w_j(s) = ||s - s_j||_2^2$.

The proof concludes by direct calculation of the 4-point condition. $\square$

*Proof of Theorem 4.2.*

Let $i = \arg\max_{j \in [N]} \sigma_j$, and decompose

$$K(s, s') = \lambda_i(s)\lambda_i(s') + R_i(s, s'), \quad R_i(s, s') := K(s, s') - \lambda_i(s)\lambda_i(s') = \sum_{j \neq i} \lambda_j(s)\lambda_j(s') \geq 0.$$

Then

$$\begin{aligned}
d(s, s') &= -\ln\left(\lambda_i(s)\lambda_i(s')\left(1 + \frac{R_i(s, s')}{\lambda_i(s)\lambda_i(s')}\right)\right) \\
&= d_i(s, s') - \varepsilon(s, s'),
\end{aligned}$$

where $\varepsilon(s, s') := \ln\left(1 + \frac{R_i(s,s')}{\lambda_i(s)\lambda_i(s')}\right) \geq 0$.

For four points $(s_x, s_y, s_z, s_w)$, define the three pair-sums under $d$ and $d_i$:

$$S_k = (k\text{-th pair-sum under } d), \quad S_k^i = (k\text{-th pair-sum under } d_i), \quad k = 1, 2, 3.$$

For example, $S_1 = d(x, y) + d(z, w)$ and $S_1^i = d_i(x, y) + d_i(z, w)$. By the decomposition,

$$S_k = S_k^i - P_k, \quad \text{where } P_k := \sum_{\text{two pairs in } S_k} \varepsilon(\cdot, \cdot), \quad P_k \in [0, 2\varepsilon_{\max}],$$

and $\varepsilon_{\max} := \max_{(u,v) \in \binom{(x,y,z,w)}{2}} \varepsilon(u, v)$.

**Claim:** $\Delta(x, y, z, w) \leq 2\varepsilon_{\max}$.

*Proof of Claim.* By Lemma E.3, $S_1^i = S_2^i = S_3^i := S$. Therefore $S_k = S - P_k$, and

$$\Delta = \underset{(1)}{\max} S_k - \underset{(2)}{\max} S_k = \underset{(1)}{\min} P_k - \underset{(2)}{\min} P_k \leq \max_k P_k - \min_k P_k \leq 2\varepsilon_{\max},$$

where the order flips because $S_k$ is decreasing in $P_k$. □

Note that now the goal is to check the condition where

$$\varepsilon(s, s') = \ln\left(1 + \frac{R_i(s, s')}{\lambda_i(s)\lambda_i(s')}\right) < \delta,$$

for some constant $\delta > 0$.

Let the centers $\{s_j\}_{j \neq i}$ be distributed as a Poisson point process $\Phi \sim \text{PPP}(\rho)$ on $[0, L]^D$ with intensity $\rho = \frac{N}{L^D}$. By the parallelogram identity, for any center $s_j$:

$$||s - s_j||_2^2 + ||s' - s_j||_2^2 = 2 ||m - s_j||_2^2 + \frac{1}{2} ||s - s'||_2^2,$$

where $m = \frac{1}{2}(s + s')$ is the midpoint. Substituting into the product of activations,

$$\lambda_j(s)\lambda_j(s') = \exp\left(-\frac{||s - s_j||_2^2 + ||s' - s_j||_2^2}{2\sigma_j^2}\right) = \underbrace{\exp\left(-\frac{||s - s'||_2^2}{4\sigma_j^2}\right)}_{\leq 1} \cdot \exp\left(-\frac{||m - s_j||_2^2}{\sigma_j^2}\right).$$

By Campbell's theorem (Theorem D.5), the expected residual satisfies

$$\mathbb{E}[R_i \mid \{\sigma_j\}] = \mathbb{E}\left[\sum_{s_j \in \Phi} \lambda_j(s)\lambda_j(s')\right]$$

$$\leq \rho \int_{\mathcal{S}} \exp\left(-\frac{||m - s_j||_2^2}{\sigma_j^2}\right) ds_j$$

$$\leq \rho \int_{\mathcal{S}} \exp\left(-\frac{||u||_2^2}{\sigma_j^2}\right) du$$

$$= \rho(\pi\sigma_j^2)^{\frac{D}{2}},$$

where the second inequality extends the domain to $\mathbb{R}^D$ (after the change of variable $u = m - s_j$), and the final equality applies the standard Gaussian integral.

Finally,

$$\mathbb{E}_{s_j}[\lambda_j(s)\lambda_j(s') \mid \sigma_j] \leq \frac{1}{L^D} \int_{\mathbb{R}^D} e^{-||m - s_j||_2^2/\sigma_j^2} ds_j = \frac{(\pi\sigma_j^2)^{\frac{D}{2}}}{L^D},$$

where $m$ is the midpoint, i.e. $m = \frac{1}{2}(s + s')$, and the second equation comes from the closed-form of the Gaussian product integral.

Summing over $j \neq i$ gives

$$\mathbb{E}[R_i] \leq (N - 1)\frac{(\pi)^{D/2}\mathbb{E}[\sigma_j^D]}{L^D} = C_D \cdot \mu,$$

where $C_D = \pi^{D/2}$, and $\mu = N\mathbb{E}[\sigma_j^D]/L^D$.

Define the event

$$\mathcal{A} := \{\sigma_i \geq cL\}, \quad c \in (0, 1).$$

By $||s - s_i|| \leq L\sqrt{D}$, on $\mathcal{A}$:

$$\lambda_i(s)\lambda_i(s') \geq e^{-\frac{D}{c^2}}. \tag{E.4}$$

By Markov's inequality (Lemma D.6):

$$\Pr[\varepsilon_{\max} \geq 2\delta \mid \mathcal{A}] \leq \frac{\mathbb{E}[\varepsilon_{\max} \mid \mathcal{A}]}{2\delta} \leq \frac{6e^{\frac{D}{c^2}}C_D\mu}{2\delta}. \tag{E.5}$$

The probability that $\mathcal{A}$ fails is given by the CDF of the exponential distribution:

$$\Pr\left[\sigma_i < cL\right] = \left(1 - e^{-\frac{cL}{\beta}}\right)^N \leq \exp\left(-Ne^{\frac{-cL}{\beta}}\right),$$

where the inequality comes from $(1 - x)^N \leq e^{-Nx}$, for $x \in [0, 1]$.

Setting the Markov bound in (E.5) equal to a target failure probability $\eta \in (0, 1)$, we obtain $2\delta = \frac{6\,e^{D/c^2}C_D\mu}{\eta}$. Finally, by a union bound over $\{\varepsilon_{\max} \geq 2\delta \mid \mathcal{A}\}$ and $\mathcal{A}^c$, with probability at least $1 - \eta - \Pr[\mathcal{A}^c]$ we have

$$\Delta \leq 2\varepsilon_{\max} \leq 2\delta = \frac{6\,e^{\frac{D}{c^2}}C_D\mu}{\eta}. \tag{E.6}$$

$\square$

### E.2.1. BREAKING CONDITIONS

**Distributions of $\sigma$.** Here we study the worst-case 4-point condition, where all 4 points sit at corners of the domain. Assume the four points $(a, b, c, d)$ form a square of side $L$ in $\mathbb{R}^D$.

We can compute the $\delta$ explicitly as follows.

**The diagonals** between opposite pairs have Euclidean distance $L\sqrt{2}$, and the kernel distance $d(s, s') \approx \frac{(L\sqrt{2})^2}{4\mathbb{E}[\sigma^2]} = \frac{2L^2}{4\mathbb{E}[\sigma^2]}$ (ignoring the constant $\ln N$). The sum of diagonals is $S_1 = \frac{4L^2}{4\mathbb{E}[\sigma^2]}$. The sides between adjacent pairs have Euclidean distance $L$, and the kernel distance $d(s, s') \approx \frac{L^2}{4\mathbb{E}[\sigma^2]}$. The sum of sides is $S_2 = \frac{2L^2}{4\mathbb{E}[\sigma^2]}$.

$\delta$ is then calculated as

$$\delta = S_1 - S_2 = \frac{L^2}{2\mathbb{E}[\sigma^2]}. \tag{E.7}$$

From (E.7) we see that any bounded distributions, such as uniform or constant-size place fields, do not preserve constant $\delta$ as $L \to \infty$.

Further, plugging other distributions into the proof of Theorem 4.2 shows that the log-normal distribution, whose tail decays more slowly than a Gaussian, also yields $\delta \to 0$. This implies that the tail behavior of the distribution affects the hyperbolicity of neural activities. ok

**Other Breaking Conditions.** Here we discuss conditions of other parameters in the system that would break statistical hyperbolicity. Specifically, breaking hyperbolicity means $\Delta$ is increasing in $L$. Based on Equation (E.6), one straightforward way to break hyperbolicity is to have

$$N = o(\frac{\mathbb{E}[\sigma^D]}{L^D}).$$

Here we interpret this as the number of place fields instead of the number of active place cells for two reasons. First, we suspect that, in the multi-field case, with each neuron having $K > 0$ place fields, the hyperbolicity of the space is controlled by the variable $NK$, which is required to scale linearly w.r.t. the size of the environment. This aligns with the experimental findings in (Harland et al., 2021). Second, ideally and theoretically, the hyperbolicity of a metric space should not depend on the dimensionality of the space, but the curvature of it, which is more closely related to how place fields cover the

environment. However, this also shows a limitation of our work, where the number of active neurons and the number of place fields are coupled. Notably, while having the constant $c \geq 1$ does not necessarily break the hyperbolicity, it is not biologically plausible to have place fields larger than the size of the environment.

### E.3. Proof of Theorem 4.8

We organize the proof as follows. In Section E.4, we state two geometric facts derived from Rauch's comparison theorem, together with the hyperbolic margin event and a Euclidean Chernoff condition on the query noise. In Section E.5, we establish a deterministic decoder bound that controls the recall error under the margin event. In Section E.6, we bridge the Euclidean and hyperbolic margins through Lemma D.2 and derive a sub-Gaussian Chernoff bound on the hyperbolic margin event. Section E.7 combines these ingredients to yield the asymptotic capacity bound, and Section E.8 isolates several regimes of interest.

### E.4. Preliminaries

Throughout this section we work on the hyperboloid model $\mathbb{H}^d_\kappa$ with curvature $\kappa < 0$, and write $\alpha := \sqrt{|\kappa|}$. We denote by $\mathbf{o}$ the origin of $\mathbb{H}^d_\kappa$, by $\mathrm{Exp}_p(\cdot)$ the exponential map at $p \in \mathbb{H}^d_\kappa$, and by $d_g(\cdot, \cdot)$ the geodesic distance. The stored patterns are $\boldsymbol{\xi}_\rho = \mathrm{Exp}_{\mathbf{o}}(x_\rho)$ for $\rho \in [M]$, with anchor points $x_\rho \in \mathbb{R}^d$, and the query is $\mathbf{v} = \mathrm{Exp}_{\mathbf{o}}(\mathrm{v})$ with $\mathrm{v} = x_\mu + \boldsymbol{\sigma}z$, where $z \sim \mathcal{N}(0, I_d)$ and $\mu \in [M]$ is the planted index.

We rely on two standard consequences of Rauch's comparison theorem (Rauch, 1951) on the simply-connected complete Hadamard manifold $\mathbb{H}^d_\kappa$.

(L1) *The inverse exponential map is globally $1$-Lipschitz*: for any $p, x, y \in \mathbb{H}^d_\kappa$,

$$\left\|\mathrm{Exp}_p^{-1}(x) - \mathrm{Exp}_p^{-1}(y)\right\|_p \leq d_g(x, y).$$

In particular, $\|\mathrm{Exp}_p^{-1}(x)\|_p = d_g(p, x)$.

(L2) *The exponential map is locally Lipschitz with explicit constant*: for any $u, w \in B(0, R) \subset T_p\mathbb{H}^d_\kappa$,

$$d_g\big(\mathrm{Exp}_p(u), \mathrm{Exp}_p(w)\big) \leq L_{\mathrm{Exp}}(R)\,\|u - w\|_p, \qquad L_{\mathrm{Exp}}(R) := \frac{\sinh(\alpha R)}{\alpha R}.$$

We next introduce the central random object of our analysis.

**Definition E.4** (Hyperbolic margin event). For a query $\mathbf{v} \in \mathbb{H}^d_\kappa$ and indices $\mu \neq \nu$, the *hyperbolic gap* is

$$\Delta^{\mathrm{H}}_{\mu\nu}(\mathbf{v}) := \langle \mathbf{v}, \boldsymbol{\xi}_\mu \rangle_{\mathscr{L}} - \langle \mathbf{v}, \boldsymbol{\xi}_\nu \rangle_{\mathscr{L}}.$$

For $\Gamma > 0$, the *margin event* at planted index $\mu$ is

$$\mathscr{M}^{\mathrm{H}}_\mu(\Gamma) := \Big\{ \min_{\nu \neq \mu} \Delta^{\mathrm{H}}_{\mu\nu}(\mathbf{v}) \geq \Gamma \Big\}. \tag{E.8}$$

The high-level strategy is to ensure that, on $\mathscr{M}^{\mathrm{H}}_\mu(\Gamma)$ with $\Gamma$ sufficiently large, the softmax weights concentrate on the planted index $\mu$, and that this margin event itself holds with high probability under the following Euclidean concentration assumption on the query noise.

**Assumption E.5** (Euclidean Chernoff condition). Fix distinct $\mu \neq \nu$, and under the conditional law $\mathrm{v} \mid \mu$ define

$$\Delta^{\mathrm{E}}_{\mu\nu}(\mathrm{v}) := \|\mathrm{v} - x_\nu\|_2^2 - \|\mathrm{v} - x_\mu\|_2^2.$$

There exist constants $\gamma_{\mathrm{E}} > 0$, $K_{\mathrm{E}} < \infty$ and $d_0 \in \mathbb{N}$ such that for all $d \geq d_0$:

(A1) $\mathbb{E}[\Delta^{\mathrm{E}}_{\mu\nu}(\mathrm{v}) \mid \mu] \geq \gamma_{\mathrm{E}} d$;

(A2) conditioned on $\mu$, the centered gap $\Delta^{\mathrm{E}}_{\mu\nu}(\mathrm{v}) - \mathbb{E}[\Delta^{\mathrm{E}}_{\mu\nu}(\mathrm{v}) \mid \mu]$ is sub-Gaussian with proxy variance at most $K_{\mathrm{E}}\boldsymbol{\sigma}^2 d$.

### E.5. Error Under Margin Event

We first show that conditional on the margin event $\mathscr{M}_\mu^{\mathrm{H}}(\Gamma)$, the softmax weights concentrate sharply on the planted index.

**Lemma E.6** (Softmax concentration). *On $\mathscr{M}_\mu^{\mathrm{H}}(\Gamma)$,*

$$\sum_{\nu \neq \mu} w_\nu(\mathbf{v}) \leq (M-1)\, e^{-\Gamma}, \qquad w_\mu(\mathbf{v}) \geq \frac{1}{1+(M-1)\, e^{-\Gamma}}. \tag{E.9}$$

*Proof.* For each $\nu \neq \mu$, $w_\nu(\mathbf{v})/w_\mu(\mathbf{v}) = \exp\big(-\Delta_{\mu\nu}^{\mathrm{H}}(\mathbf{v})\big) \leq e^{-\Gamma}$ on $\mathscr{M}_\mu^{\mathrm{H}}(\Gamma)$. Summing over $\nu \neq \mu$ and using $\sum_\rho w_\rho(\mathbf{v}) = 1$ yields both inequalities. $\qquad\square$

We now translate softmax concentration into a deterministic recall guarantee.

**Proposition E.7** (Decoder bound). *Set $u_\rho := \mathrm{Exp}_{\mathbf{v}}^{-1}(\boldsymbol{\xi}_\rho) \in T_{\mathbf{v}}\mathbb{H}_\kappa^d$ and $\bar{u} := \sum_\rho w_\rho(\mathbf{v})\, u_\rho$, so that $H(\mathbf{v}) = \mathrm{Exp}_{\mathbf{v}}(\bar{u})$. Define $R(\mathbf{v}) := \max_\rho \|u_\rho\|_{\mathbf{v}} = \max_\rho d_g(\mathbf{v}, \boldsymbol{\xi}_\rho)$, where the second equality follows from* (L1). *Then on $\mathscr{M}_\mu^{\mathrm{H}}(\Gamma)$,*

$$d_g\big(H(\mathbf{v}), \boldsymbol{\xi}_\mu\big) \leq 2\, r_{\max}\, L_{\mathrm{Exp}}\big(R(\mathbf{v})\big)\, (M-1)\, e^{-\Gamma}. \tag{E.10}$$

*Proof.* By (L2) applied with $u = \bar{u}$ and $w = u_\mu \in B(0, R(\mathbf{v}))$,

$$d_g\big(H(\mathbf{v}), \boldsymbol{\xi}_\mu\big) = d_g\big(\mathrm{Exp}_{\mathbf{v}}(\bar{u}), \mathrm{Exp}_{\mathbf{v}}(u_\mu)\big) \leq L_{\mathrm{Exp}}\big(R(\mathbf{v})\big)\, \|\bar{u} - u_\mu\|_{\mathbf{v}}. \tag{E.11}$$

By the triangle inequality and (L1) applied to each pair $(\boldsymbol{\xi}_\nu, \boldsymbol{\xi}_\mu)$,

$$\|\bar{u} - u_\mu\|_{\mathbf{v}} = \Big\|\sum_{\nu \neq \mu} w_\nu(\mathbf{v})\, (u_\nu - u_\mu)\Big\|_{\mathbf{v}} \leq \sum_{\nu \neq \mu} w_\nu(\mathbf{v})\, d_g(\boldsymbol{\xi}_\nu, \boldsymbol{\xi}_\mu) \leq 2\, r_{\max} \sum_{\nu \neq \mu} w_\nu(\mathbf{v}).$$

Combining with (E.11) and Lemma E.6 yields (E.10). $\qquad\square$

*Remark* E.8. The Lipschitz constant $L_{\mathrm{Exp}}(R(\mathbf{v}))$ appears only as a multiplicative prefactor in (E.10), and will therefore enter the capacity bound only through its logarithm. In the large-radius regime $\alpha r_{\max} \gg 1$, one has $\log L_{\mathrm{Exp}}(R(\mathbf{v})) = \Theta(\alpha r_{\max})$, which is of strictly lower order than the leading $d\, f_\kappa(r_{\min})$ term appearing below.

### E.6. Chernoff Bound on the Margin Event

To control the margin event probabilistically, we first restrict to a high-probability set on which the radial norms are well-behaved.

**Lemma E.9** (Noise control). *Let $\mathscr{E}_z := \{\|z\|_2 \leq \sqrt{2d}\}$. Then $\mathrm{Pr}[\mathscr{E}_z^c] \leq e^{-d/2}$, and on $\mathscr{E}_z$, $\|\mathbf{v}\|_2 \in [r_{\min}^*, r_{\max}^*]$ with*

$$r_{\min}^* := \max\{r_{\min} - \boldsymbol{\sigma}\sqrt{2d},\, 0\}, \qquad r_{\max}^* := r_{\max} + \boldsymbol{\sigma}\sqrt{2d}.$$

*Moreover, under the noise condition*

$$\boldsymbol{\sigma}\sqrt{2d} \leq r_{\min}/2, \tag{E.12}$$

*we have $r_{\min}^* \geq r_{\min}/2 > 0$ and $R^* := r_{\max}^* + r_{\max} \leq 5r_{\max}/2$, so that $L_{\mathrm{Exp}}(R(\mathbf{v})) \leq L_{\mathrm{Exp}}(R^*)$ holds deterministically on $\mathscr{E}_z$.*

*Proof.* The Gaussian-norm bound $\mathrm{Pr}[\|z\|_2 > \sqrt{2d}] \leq e^{-d/2}$ is a standard $\chi^2$ tail bound (Wainwright, 2019). The remaining claims follow by the triangle inequality and from $g(r) = \sinh(\alpha r)/(\alpha r)$ being monotone increasing. $\qquad\square$

We now combine Assumption E.5 with the Euclidean-to-hyperbolic bridge of Lemma D.2 to obtain a Chernoff bound on the margin event.

**Proposition E.10** (Hyperbolic Chernoff). *Suppose Assumption E.5 and the noise bound (E.12) hold. Define the inflated curvature parameters*

$$\gamma_\kappa^H := f_\kappa(r_{\min}^*)\,\gamma^E - \widetilde{\mathsf{Pen}}_\kappa(r_{\min}^*, r_{\max}^*)/d, \qquad K_\kappa^H := f_\kappa(r_{\min}^*)^2\,K^E\,\sigma^2. \tag{E.13}$$

*Then for any $\Gamma < \gamma_\kappa^H d$,*

$$\Pr\!\big[\mathscr{M}_\mu^H(\Gamma)^c \mid \mu,\, \mathscr{E}_z\big] \;\le\; (M-1)\,\exp\!\left(-\frac{(\gamma_\kappa^H d - \Gamma)^2}{2K_\kappa^H d}\right). \tag{E.14}$$

*Proof.* On $\mathscr{E}_z$, by Lemma D.2 applied with the inflated radii $[r_{\min}^*, r_{\max}^*]$,

$$\Delta_{\mu\nu}^H(\mathbf{v}) \;\ge\; f_\kappa(r_{\min}^*)\,\Delta_{\mu\nu}^E(\mathbf{v}) \;-\; \widetilde{\mathsf{Pen}}_\kappa(r_{\min}^*, r_{\max}^*). \tag{E.15}$$

Define the (sub-Gaussian) surrogate

$$\widetilde{\Delta}_{\mu\nu}^H \;:=\; f_\kappa(r_{\min}^*)\,\Delta_{\mu\nu}^E(\mathbf{v}) \;-\; \widetilde{\mathsf{Pen}}_\kappa(r_{\min}^*, r_{\max}^*),$$

so that $\Delta_{\mu\nu}^H \ge \widetilde{\Delta}_{\mu\nu}^H$ on $\mathscr{E}_z$. Because $\widetilde{\Delta}_{\mu\nu}^H$ is a deterministic affine function of $\Delta_{\mu\nu}^E(\mathbf{v})$, Assumption E.5 yields

$$\mathbb{E}\big[\widetilde{\Delta}_{\mu\nu}^H \mid \mu\big] \;\ge\; f_\kappa(r_{\min}^*)\,\gamma^E\,d - \widetilde{\mathsf{Pen}}_\kappa(r_{\min}^*, r_{\max}^*) \;=\; \gamma_\kappa^H\,d,$$

and $\widetilde{\Delta}_{\mu\nu}^H - \mathbb{E}\big[\widetilde{\Delta}_{\mu\nu}^H \mid \mu\big]$ is sub-Gaussian with proxy variance $K_\kappa^H d$. The standard one-sided sub-Gaussian Chernoff bound gives, for any $\Gamma < \gamma_\kappa^H d$,

$$\Pr\!\big[\widetilde{\Delta}_{\mu\nu}^H < \Gamma \mid \mu\big] \;\le\; \exp\!\left(-\frac{(\gamma_\kappa^H d - \Gamma)^2}{2K_\kappa^H d}\right).$$

A union bound over $\nu \ne \mu$, combined with $\Delta_{\mu\nu}^H \ge \widetilde{\Delta}_{\mu\nu}^H$ on $\mathscr{E}_z$, yields (E.14). $\qquad\square$

### E.7. Capacity Bound

We now combine the above two steps to control the total recall error. We first present a more general result below (Theorem E.11), where Theorem 4.8 is a direct corollary.

**Theorem E.11** (Capacity). *Suppose Assumption E.5 and the noise bound (E.12) hold, that $\widetilde{\mathsf{Pen}}_\kappa(r_{\min}^*, r_{\max}^*) = o(d\,f_\kappa(r_{\min}))$, and that the SNR condition*

$$\sigma^2\,f_\kappa(r_{\min}^*) \;=\; O(1) \quad\Longleftrightarrow\quad \sigma \;=\; O\!\Big(\sqrt{|\kappa|}\,r_{\min}\,e^{-\alpha r_{\min}}\Big)z \tag{E.16}$$

*holds. Then, as $d \to \infty$, the admissible number of stored patterns satisfies*

$$\log M \;=\; \Theta\big(d\,f_\kappa(r_{\min})\big) \;-\; \widetilde{\mathsf{Pen}}_\kappa(r_{\min}, r_{\max}) \;=\; \Theta\!\left(\frac{d}{|\kappa|}\left(\frac{\sinh(\alpha r_{\min})}{r_{\min}}\right)^2\right) \;-\; \widetilde{\mathsf{Pen}}_\kappa. \tag{E.17}$$

*Proof.* By Proposition E.7, on $\mathscr{M}_\mu^H(\Gamma) \cap \mathscr{E}_z$ the recall succeeds (i.e. $d_g(H(\mathbf{v}), \boldsymbol{\xi}_\mu) \le \varepsilon$) provided

$$\Gamma \;\ge\; \Gamma_\star(M) \;:=\; \log\!\left(\frac{2\,r_{\max}\,L_{\mathrm{Exp}}(R^*)\,(M-1)}{\varepsilon}\right) \;=\; \log M + O(\alpha r_{\max}). \tag{E.18}$$

A union bound over recall failure events gives

$$\Pr[\text{recall fail}] \;\le\; e^{-d/2} \;+\; (M-1)\,\exp\!\left(-\frac{(\gamma_\kappa^H d - \Gamma_\star(M))^2}{2K_\kappa^H d}\right). \tag{E.19}$$

Set $\log M = c\,\gamma_\kappa^H d$ for some $c \in (0,1)$ to be chosen. Then $\Gamma_\star(M) = \log M + O(\alpha r_{\max})$, and the second term in (E.19) has exponent

$$\frac{(1-c)^2(\gamma_\kappa^H)^2 d}{2K_\kappa^H}(1+o(1)) \;=\; \frac{(1-c)^2}{2}\,\frac{\gamma_\kappa^H}{K_\kappa^H}\,\gamma_\kappa^H d\,(1+o(1)).$$

Vanishing total error therefore requires

$$c < \frac{(1-c)^2}{2} \frac{\gamma_\kappa^{\mathrm{H}}}{K_\kappa^{\mathrm{H}}}, \tag{E.20}$$

which admits a positive-constant solution $c$ if and only if $\gamma_\kappa^{\mathrm{H}}/K_\kappa^{\mathrm{H}} = \Omega(1)$. By (E.13), this ratio is $\gamma_\kappa^{\mathrm{H}}/K_\kappa^{\mathrm{H}} \sim \gamma^{\mathrm{E}}/(f_\kappa(r_{\min}^*)K^{\mathrm{E}}\boldsymbol{\sigma}^2)$, which is $\Omega(1)$ precisely under the SNR condition (E.16). Substituting $\gamma_\kappa^{\mathrm{H}} \sim f_\kappa(r_{\min})\gamma^{\mathrm{E}}$ (using $\widetilde{\mathrm{Pen}}_\kappa = o(d\, f_\kappa(r_{\min}))$) into $\log M = c\,\gamma_\kappa^{\mathrm{H}}d$ yields (E.17). $\qquad\square$

*Remark* E.12. The Lipschitz constant $L_{\mathrm{Exp}}(R^*)$ contributes only an additive $O(\alpha r_{\max})$ term to $\Gamma_\star(M)$, which is dominated by the leading $d\, f_\kappa(r_{\min})$ term. Equivalently, the prefactor $e^{\alpha R^*}$ from $L_{\mathrm{Exp}}$ enters the capacity bound only after taking a logarithm: an exponentially-bad multiplicative factor in the recall error contracts to a linear additive cost in $\log M$. This is the key reason why the double-exponential scaling in $r_{\min}$ is preserved.

**Corollary E.13.** *Let $\kappa < 0$ and $\alpha = \sqrt{|\kappa|}$. Suppose Assumption E.5 and the noise bound (E.12) hold, that the stored patterns lie in a narrow shell with $\alpha r_{\min} \gg 1$ and*

$$\Delta_r = o\left(\frac{d}{\alpha\, r_{\min}^2}\right), \tag{E.21}$$

*and that the SNR condition*

$$\boldsymbol{\sigma} = O\left(\sqrt{|\kappa|}\, r_{\min}\, e^{-\alpha r_{\min}}\right) \tag{E.22}$$

*holds. Then, as $d \to \infty$, the admissible number of stored patterns satisfies*

$$\log M = \Theta\left(\frac{d}{|\kappa|} \frac{e^{2\alpha r_{\min}}}{r_{\min}^2}\right), \qquad M = \exp\left(\Theta\left(\frac{d}{|\kappa|} \frac{e^{2\alpha r_{\min}}}{r_{\min}^2}\right)\right). \tag{E.23}$$

*Proof.* The width condition (E.21) ensures $\widetilde{\mathrm{Pen}}_\kappa(r_{\min}, r_{\max}) = o(d\, f_\kappa(r_{\min}))$ by Corollary E.17, so the hypotheses of Theorem E.11 are satisfied. In the large-radius regime $\alpha r_{\min} \gg 1$,

$$f_\kappa(r_{\min}) = \frac{1}{2|\kappa|}\left(\frac{\sinh(\alpha r_{\min})}{r_{\min}}\right)^2 = \Theta\left(\frac{1}{|\kappa|} \frac{e^{2\alpha r_{\min}}}{r_{\min}^2}\right).$$

Substituting into (E.17) and absorbing the penalty into the lower-order term yields (E.23). $\qquad\square$

*Remark* E.14. We refer to (E.23) as *double-exponential capacity*: $M$ is exponential in $d$, and the rate of that exponential is itself exponential in the radius $r_{\min}$. This radial amplification is a strictly hyperbolic phenomenon, with no analogue in Euclidean associative memory models (Ramsauer et al., 2020; Krotov & Hopfield, 2021), where capacity scales at most exponentially in $d$ with a curvature-independent rate.

*Remark* E.15. Corollary E.13 clarifies the role of each hypothesis. The width condition (E.21) controls the bridge penalty so that the hyperbolic margin inherits the sub-Gaussian Chernoff exponent of the Euclidean gap. The SNR condition (E.22) ensures that the variance proxy $K_\kappa^{\mathrm{H}} = f_\kappa(r_{\min}^*)^2 K^{\mathrm{E}}\boldsymbol{\sigma}^2$ does not absorb the entire amplification. Together, they pin the capacity rate to $f_\kappa(r_{\min})$, which carries the double-exponential growth.

### E.8. Capacity Regimes

The capacity rate (E.17) admits several natural regimes, summarized in Table 3.

We now isolate the two regimes of primary interest.

**Corollary E.16** (Thin shell). *Suppose $r_{\min} = r_{\max} = r$, so that $\Delta_r = 0$ and $\widetilde{\mathrm{Pen}}_\kappa = 0$. Then under the conditions of Theorem E.11,*

$$\log M = \Theta\left(\frac{d}{|\kappa|}\left(\frac{\sinh(\alpha r)}{r}\right)^2\right). \tag{E.24}$$

*In the small-radius regime $\alpha r \ll 1$, this reduces to $\log M = \Theta(d)$. In the large-radius regime $\alpha r \gg 1$,*

$$\log M = \Theta\left(\frac{d}{|\kappa|} \frac{e^{2\alpha r}}{r^2}\right),$$

*exhibiting double-exponential radial amplification: $M = \exp\left(\Theta(d\, e^{2\alpha r}/(|\kappa| r^2))\right)$.*

*Table 3.* Capacity scaling regimes under the hyperbolic Chernoff analysis. Here $\alpha = \sqrt{|\kappa|}$, $r = r_{\min} = r_{\max}$ in the thin-shell regime, and $\Delta_r = r_{\max} - r_{\min}$ otherwise.

| Regime | $\Delta_r$ | $\widetilde{\mathrm{Pen}}_\kappa$ | $\log M$ |
|---|---|---|---|
| Thin shell | $0$ | $0$ | $\Theta\big(d\, f_\kappa(r)\big) = \Theta\big(\frac{d}{|\kappa|} \frac{e^{2\alpha r}}{r^2}\big)$ |
| Narrow shell | $O(1/d)$ | $O(1)$ | $\Theta\big(d\, f_\kappa(r_{\min})\big)$ |
| Sub-critical shell | $o\big(\frac{d}{\alpha r_{\min}^2}\big)$ | $o(d\, f_\kappa)$ | $\Theta\big(d\, f_\kappa(r_{\min})\big)$ |
| Super-critical shell | $\Omega\big(\frac{d}{\alpha r_{\min}^2}\big)$ | $\Omega(d\, f_\kappa)$ | **Collapse** |

**Corollary E.17** (Width–amplification tradeoff). *Exponential-in-$d$ capacity persists if and only if $\widetilde{\mathrm{Pen}}_\kappa(r_{\min}, r_{\max}) = o(d\, f_\kappa(r_{\min}))$. In the large-radius regime $\alpha r_{\min} \gg 1$, this condition is implied by*

$$\Delta_r = o\left(\frac{d}{\alpha\, r_{\min}^2}\right), \tag{E.25}$$

*showing that admissible radial spread is at most of order $d/(\alpha r_{\min}^2)$ before the penalty term dominates and capacity collapses.*

*Remark* E.18. The SNR condition (E.16) admits a natural interpretation as a noise-to-curvature balance: in order for the hyperbolic amplification factor $f_\kappa$ to transfer fully into capacity, the noise $\sigma$ must shrink at most inverse-exponentially with $\alpha r_{\min}$. Combined with the milder condition (E.12), this is the binding constraint as $\alpha r_{\min} \to \infty$.

### E.9. Derivations

#### E.9.1. DERIVATION OF POSTERIOR

By Bayes' rule, the posterior over memory index $\mu$ is

$$p(\mu \mid \boldsymbol{n}) = \frac{p(\boldsymbol{n} \mid \mu)\, p(\mu)}{p(\boldsymbol{n})}.$$

Taking logarithms,

$$\log p(\mu \mid \boldsymbol{n}) = \log p(\boldsymbol{n} \mid \mu) + \log p(\mu) - \log p(\boldsymbol{n}).$$

Assuming conditional independence of the spike counts across neurons given $\mu$, the Poisson log-likelihood gives

$$\log p(\boldsymbol{n} \mid \mu) = \sum_{i=1}^{N} \big(n_i \log \lambda_i^\mu - \lambda_i^\mu \Delta t\big) + C_1,$$

where $C_1$ collects terms (e.g. factorials $\log(n_i!)$) that do not depend on $\mu$ when the rates $\{\lambda_i^\mu\}$ are fixed. Hence

$$\log p(\mu \mid \boldsymbol{n}) = \sum_{i=1}^{N} \big(n_i \log \lambda_i^\mu - \lambda_i^\mu \Delta t\big) + \log p(\mu) - \log p(\boldsymbol{n}) + C_1.$$

Equivalently, absorbing the $\mu$-independent factor $e^{C_1}/p(\boldsymbol{n})$ into normalization,

$$p(\mu \mid \boldsymbol{n}) \propto \exp\Big(\sum_{i=1}^{N} \big(n_i \log \lambda_i^\mu - \lambda_i^\mu \Delta t\big) + \log p(\mu)\Big).$$

#### E.9.2. DERIVATION OF (2.6)

*Proof.* Write the log-posterior scores

$$h_\mu(\boldsymbol{n}) := \log p(\boldsymbol{n} \mid \mu) + \log p(\mu),$$

so that $\log p(\mu \mid \boldsymbol{n}) = h_\mu(\boldsymbol{n}) - \log p(\boldsymbol{n})$. Then

$$
\begin{aligned}
\exp\big(\log p(\mu \mid \boldsymbol{n})\big) &= \exp\big(h_\mu(\boldsymbol{n}) - \log p(\boldsymbol{n})\big) \\
&= \exp\big(h_\mu(\boldsymbol{n})\big)\,\exp\big(-\log p(\boldsymbol{n})\big) \\
&= \frac{\exp\big(h_\mu(\boldsymbol{n})\big)}{\exp\big(\log p(\boldsymbol{n})\big)} \\
&= \frac{\exp\big(h_\mu(\boldsymbol{n})\big)}{p(\boldsymbol{n})}.
\end{aligned}
$$

Summing over all $\mu$ and using $\sum_\nu p(\nu \mid \boldsymbol{n}) = 1$,

$$
1 = \sum_\nu \frac{\exp\big(h_\nu(\boldsymbol{n})\big)}{p(\boldsymbol{n})},
$$

so $p(\boldsymbol{n}) = \sum_\nu \exp(h_\nu(\boldsymbol{n}))$. Finally, let $\boldsymbol{h} := \big(h_1(\boldsymbol{n}), \ldots, h_M(\boldsymbol{n})\big)$. Then

$$
p(\mu \mid \boldsymbol{n}) = \operatorname{softmax}_\mu(\boldsymbol{h}) = \frac{\exp\big(h_\mu(\boldsymbol{n})\big)}{\sum_{\nu=1}^{M} \exp\big(h_\nu(\boldsymbol{n})\big)}.
$$

$\square$

# F. Simulation Details

## F.1. Pattern Completion

**Baseline Implementation.** We implement two associative memory models, the modern Hopfield networks (Ramsauer et al., 2020) and dense associative memory (Krotov & Hopfield, 2021). For MHN, its update rule is

$$v^{(\text{new})} \leftarrow \sum_{\mu=1}^{M} \text{softmax}_\mu \left( \beta v^\top \xi_1, \cdots, \beta v^\top \xi_M \right).$$

For DAM, its update rule is

$$v^{(\text{new})} \leftarrow \sum_{\mu=1}^{M} w_\mu \cdot \xi_\mu, \quad w_\mu = (\beta v^\top \xi_\mu)^n.$$

To compute the posterior for DAM, we have

$$p(\mu \mid v) = \frac{w_\mu}{\sum_\nu w_\nu}.$$

**Sampling.** For synthetic memory patterns, we uniformly sample them inside a ball with radius $r_{\max}$. To do so, we adopt a common approach by sampling the point direction and radius separately.

**Preprocessing.** For data preprocessing, we only use `torch.ToTensor` to normalize pixels to $[0, 1]$. Next, for real-world images, we apply PCA to reduce their dimensionality to some $d$, we then rescale the memory patterns with $r_{\max}$. For KFM, we project them to the hyperboloid model by first concatenating an extra dimension to the first entry, with value 1 (totangent), then apply the exponential map.

**Hyperparameters.** The hyperparameters are listed in Table 4. Specifically, the major difference between the synthetic data and real-world data is we use $\beta = 1.0$ for synthetic data, and $\beta = 10.0$ for real-world data. For all results, we report the mean and standard deviation across 10 runs.

*Table 4.* Hyperparameter used in the pattern completion task.

| parameter | Value | | |
|---|---|---|---|
| **Dataset** | MNIST | CIFAR10 | Synthetic |
| max. update steps | 64 | 64 | 64 |
| $M$ | $[10, 1000]$ | $[10, 1000]$ | $[10, 1000]$ |
| $d$ | $\{10, 20, 100\}$ | $\{10, 20, 100\}$ | $\{10, 20, 100\}$ |
| $\beta$ | 1 | 10 | 10 |
| $\epsilon$ | 0.01 | 0.01 | 0.01 |
| $r_{\max}$ | 3 | 3 | 3 |
| DAM order | 10 | 10 | 10 |

## F.2. Multiple Instance Learning

Here we briefly review the concept of multiple instance learning (MIL). MIL is a variant of supervised learning in which the training set consists of labeled bags containing multiple instances. The objective is to predict bag-level labels from the instances within each bag, making MIL particularly suitable for settings where instance-level annotation is difficult or impractical but bag-level labels are available. Applications include medical imaging, where bags correspond to images and instances to image patches, and document classification, where bags correspond to documents and instances to words or sentences. The statistics of the benchmark dataset we used in the paper is in Table 5.

*Table 5.* Statistics of MIL benchmark datasets

| Name | Instances | Features | Bags | +bags | −bags |
|------|-----------|----------|------|-------|-------|
| Elephant | 1391 | 230 | 200 | 100 | 100 |
| Fox | 1302 | 230 | 200 | 100 | 100 |
| Tiger | 1220 | 230 | 200 | 100 | 100 |

## F.3. Machine Learning Layers

**Layer Definition**   The hyperbolic attention layer maps Euclidean inputs to a Hadamard manifold $\mathcal{M} = \mathbb{H}^d_\kappa$ for associative recall and returns the Euclidean mappings. Let $R \in \mathbb{R}^{S \times d}$ and $Y \in \mathbb{R}^{M \times d}$ denote the state and memory matrices, with learned weights $W_Q, W_K, W_V \in \mathbb{R}^{d \times d}$. Inputs are projected into $\mathcal{M}$ via the exponential map $\mathrm{Exp}_o$ at the origin $\boldsymbol{o} = [\sqrt{k}, 0]$:

$$\begin{aligned}
\boldsymbol{Q} &= \mathrm{Exp}_o(RW_Q), \\
\boldsymbol{K} &= \mathrm{Exp}_o(YW_K), \\
\boldsymbol{V} &= \mathrm{Exp}_o(YW_KW_V).
\end{aligned} \tag{F.1}$$

The attention weights are determined by the Minkowski inner product $\langle \boldsymbol{q}_i, \boldsymbol{k}_j \rangle_L$ such that $\alpha_{ij} = \mathrm{softmax}_j\left(-\beta\langle \boldsymbol{q}_i, \boldsymbol{k}_j \rangle_L\right)$.

Then, the manifold output $\boldsymbol{Z} \in \mathcal{M}^{S \times d}$ is computed via Karcher flow, approximating the weighted Fréchet Mean of $\boldsymbol{V}$ through the update:

$$\boldsymbol{z}_i = \mathrm{Exp}_{\boldsymbol{q}_i}\left(\sum_{j=1}^{M} \alpha_{ij}\mathrm{Exp}_{\boldsymbol{q}_i}^{-1}(\boldsymbol{v}_j)\right). \tag{F.2}$$

where $\boldsymbol{Z} = [\boldsymbol{z}_1, \boldsymbol{z}_2, \cdots, \boldsymbol{z}_S]$.

**Types of layers**   The general hyperbolic attention mechanism described above can be specialized into three distinct layer types by specifying different sources of the state $R$ and memory $Y$. These layers are the hyperbolic analogs of the `Hopfield`, `HopfieldPooling`, and `HopfieldLayer` modules described in (Ramsauer et al., 2020).

**Karcher Flow Attention:** This is the implentation of (F.2). Both $R$ and $Y$ are dynamic inputs from preceding layers or another input source. This configuration performs hyperbolic self-attention or cross-attention between two sets of vectors.

**Karcher Flow Pooling:** In this layer, patterns propagate via the memory patterns $Y$. The stored memories $Y$ are summarized through queries by the static patterns $\Xi \in \mathbb{R}^{S \times d}$. We define $\alpha_{ij} = \mathrm{softmax}_j\left(-\beta\langle \boldsymbol{\xi}_i, \boldsymbol{k}_j \rangle_L\right)$ and the update rule for this layer is given by:

$$\boldsymbol{z}_i = \mathrm{Exp}_{\boldsymbol{\xi}_i}\left(\sum_{j=1}^{M} \alpha_{ij}\mathrm{Exp}_{\boldsymbol{\xi}_i}^{-1}(\boldsymbol{v}_j)\right). \tag{F.3}$$

where $\boldsymbol{\Xi} = \mathrm{Exp}_{\boldsymbol{o}}(\Xi)$.

**Karcher Flow Layer:** In this layer, patterns propagate via the state patterns $R$. Memories are fixed and represented by the weight matrix $W_K \in \mathbb{R}^{M \times d}$. The input $R$ acts as the query set. We define $\alpha_{ij} = \mathrm{softmax}_j\left(-\beta\langle \boldsymbol{r}_i, (\boldsymbol{W}_K)_j \rangle_L\right)$ and the update rule for this layer is given by:

$$\boldsymbol{z}_i = \mathrm{Exp}_{\boldsymbol{r}_i}\left(\sum_{j=1}^{M} \alpha_{ij}\mathrm{Exp}_{\boldsymbol{r}_i}^{-1}((\boldsymbol{W}_V)_j)\right). \tag{F.4}$$

where $\boldsymbol{R} = \mathrm{Exp}_{\boldsymbol{o}}(R)$, $\boldsymbol{W}_K = \mathrm{Exp}_{\boldsymbol{o}}(W_K)$, and $\boldsymbol{W}_V = \mathrm{Exp}_{\boldsymbol{o}}(W_V)$.

*Table 6.* Hyperparameters used in attention tasks.

| Parameter | MNIST | MIL |
|---|---|---|
| Optimizer | AdamW | AdamW |
| Learning Iteration $N$ | 14 | 100 |
| Batch Size | 64 | 16 |
| Update Rule Iteration | 1 | 1 |
| Learning Rate | 0.001 | 0.001 |
| Learning Rate Decay ($\gamma$) | 0.96 | 0.96 |
| LR Scheduler | Step Decay | Step Decay |
| Hidden Dimension $d$ | $\{4, 8, 32\}$ | 128 |
| Scaling Factor ($\beta$) | $\frac{1}{\sqrt{d}}$ | $\frac{1}{\sqrt{128}}$ |
| Bag Dropout | — | 0.5 |

# G. Additional Simulations

## G.1. Pattern Completion

Here we further investigate the case of how increasing $r_{\max}$ would help memory storage (pattern separation). Specifically, we now evaluate this property on real world datasets. We follow the style in Figure 2(b) and plot the model performance under different values of $r_{\max}$ in the same subplot. We observe that not only did KFM performs the best when $d = 10$. We can also observe that, on the tail of the performance curve (when $M$ is large), only KFM is able to gain notable performance boost. Meanwhile, other baseline models still shows a rapid recall rate decay when $M$ is close to 1000.

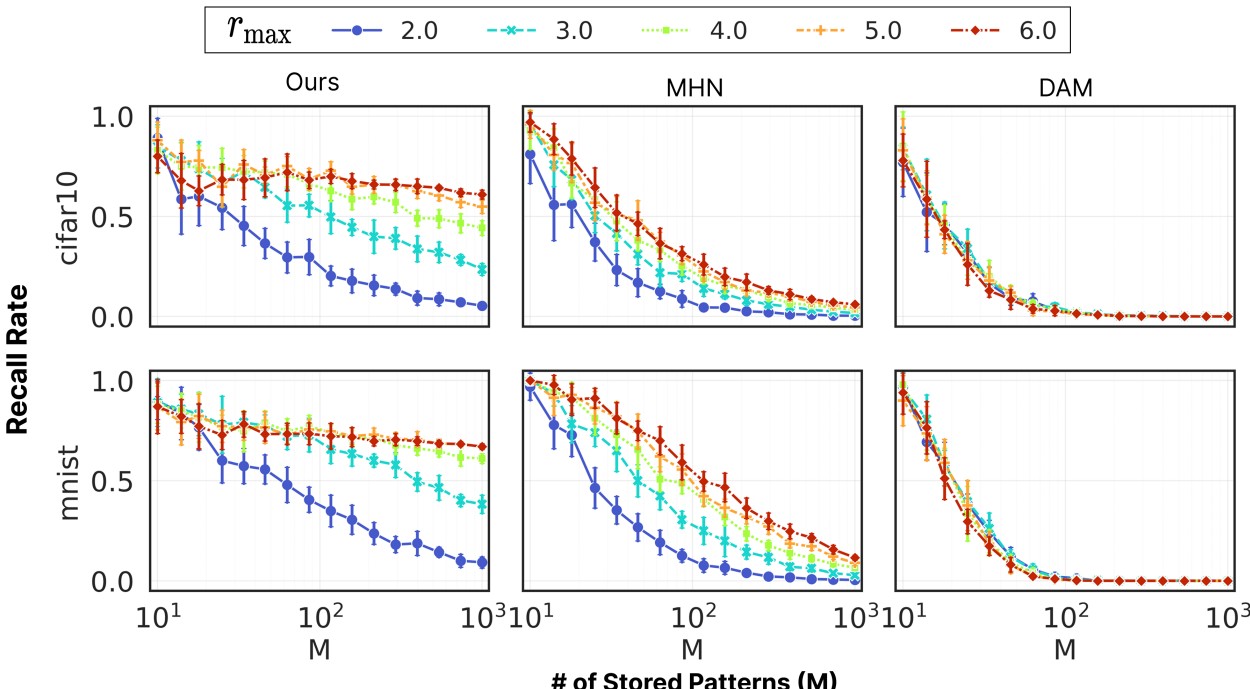

*Figure 4.* **Recall success rate under different values of $r_{\max}$. Columns left-to-right: KFM, MHN, DAM.** To further validate our double-exponential capacity result in $r_{\max}$, we perform pattern completion on different datasets, rescaling the images to different $r_{\max}$. We observe that KFM achieves the best recall rate when $M$ is large. In particular, at $M = 1000$, KFM is the only model with substantial capacity improvement. Here we set the PCA dimension to 10 for all models and datasets.

## G.2. Hyperbolicity Simulation

Here we empirically compute the hyperbolicity constant $\delta$ when the neural representation is generated from different tuning curve families. Specifically, we define 5 tuning curves under different place field size distributions, and see how their empirical $\delta$ scales with respect to the size of the environment. We define their probability density functions and mean values as follows. Specifically, we choose the parameters carefully so that all distributions has the same mean values. The results are in Figure 5. We observe that both the exponential distribution and the log-normal distributions remain statistically hyperbolic when $L$ increases, which are the two distributions reported in (Zhang et al., 2023). Both constant and uniform distributions are not able to remain statistically hyperbolic as their delta grows significantly with $L$. Interestingly, the half-normal distribution is able to also maintain low $\delta$, but is slightly less hyperbolic than the exponential and log-normal distributions.

**Exponential.**

$$X \sim \text{Exp}(\lambda = 1), \quad f_X(x) = \begin{cases} e^{-x}, & x \geq 0 \\ 0, & x < 0 \end{cases}, \quad \mathbb{E}[X] = 1.$$

**Log-normal.**

$$X \sim \text{LogNormal}(\mu = -0.5, \sigma = 1), \quad f_X(x) = \frac{1}{x\sqrt{2\pi}} \exp\left(-\frac{(\ln x + 0.5)^2}{2}\right), \qquad x > 0, \quad \mathbb{E}[X] = e^{\mu + \sigma^2/2} = 1.$$

**Uniform.**

$$X \sim \text{Uniform}(0.2, 1.8), \quad f_X(x) = \begin{cases} \frac{1}{1.6}, & 0.2 \leq x \leq 1.8 \\ 0, & \text{otherwise} \end{cases}, \quad \mathbb{E}[X] = \frac{0.2 + 1.8}{2} = 1.$$

**Constant.**

$$X = 1, \quad f_X(x) = Dirac(x-1),$$

where $Dirac$ denotes the Dirac delta distribution.

**Half-normal.**

$$X \sim \text{HalfNormal}\left(\sigma = \sqrt{\frac{\pi}{2}}\right), \quad f_X(x) = \sqrt{\frac{2}{\pi\sigma^2}} \exp\left(-\frac{x^2}{2\sigma^2}\right), \qquad x \geq 0, \quad \sigma = \sqrt{\frac{\pi}{2}}, \quad \mathbb{E}[X] = \sigma\sqrt{\frac{2}{\pi}} = 1.$$

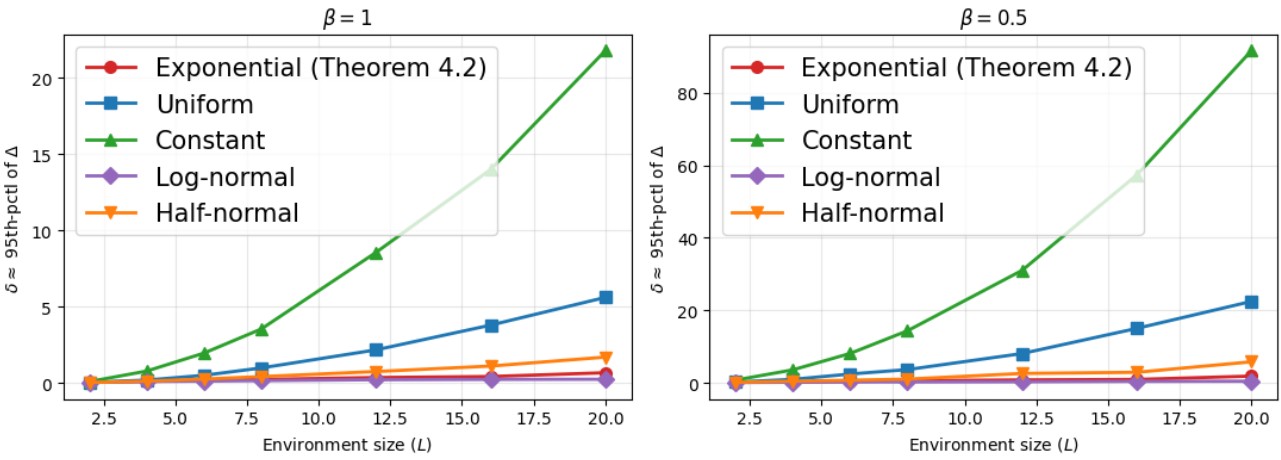

*Figure 5.* **Empirical hyperbolicity under different place field size distributions.** We observe that both the exponential distribution and the log-normal distributions remain statistically hyperbolic when $L$ increases, which are the two distributions reported in (Zhang et al., 2023). Both constant and uniform distributions are not able to remain statistically hyperbolic as their delta grows significantly with $L$. Interestingly, the half-normal distribution is able to also maintain low $\delta$, but is slightly less hyperbolic than the exponential and log-normal distributions.

