# OpenReview forum: "Hyperbolic neural population geometry benefits computation"
_ICML.cc/2026/Conference — ICML 2026 regular_

### Official Review · Reviewer_FPK9 · 2026-03-03

**Soundness:** 2
**Presentation:** 2
**Significance:** 2
**Originality:** 2
**Overall Recommendation:** 4
**Confidence:** 3

**Summary:**

This paper aims to establish a theoretical framework for the neurobiological phenomenon that "neural population activity induces hyperbolic geometry", and on this basis, design a novel machine learning associative memory model with ultra-large storage capacity. The core logical chain of the paper can be decomposed into the following four levels:

- Hyperbolic Metric Modeling
This paper completes the full modeling workflow from biological phenomena to hyperbolic metrics. Grounded in the latest empirical findings from the hippocampal CA1 region, the paper puts forward the core hypothesis that the widths of neuronal place fields follow an exponential distribution. On the dimension of metric induction, through the mathematical derivation in Theorem 4.1, the paper proves that as the number of neurons tends to infinity, Gaussian tuning curves constructed based on such exponentially distributed receptive fields statistically induce an approximately tree-like semimetric space satisfying the δ-hyperbolic property in the stimulus space.

- Mapping Associative Memory to Bayesian Inference
This paper builds a theoretical bridge between associative memory and Bayesian inference. It establishes a key equivalence relation, proving that the recall dynamics of Modern Hopfield Networks (MHN) can approximately compute the Bayesian optimal estimator in the sense of minimum mean squared error (MMSE). Based on this theoretical perspective, the paper equates pattern retrieval in associative memory to probabilistic decoding inference on a manifold, which lays a core foundation for the subsequent reconstruction of the network architecture in non-Euclidean geometric spaces.

- Algorithmic Innovation
Combining the theoretical insights from the previous two parts, this paper completes the algorithmic innovation of the associative memory model in hyperbolic space. In terms of update rule design, the paper proposes an associative memory model that operates directly in hyperbolic space (adopting the Lorentz/hyperboloid model \mathbb{H}_\kappa^d), with its state updates implemented via Karcher flow. In terms of leveraging metric advantages, the model proposed in the paper adopts the Lorentz (Minkowski) inner product to encode hyperbolic distances, which has exactly the same computational complexity as the Euclidean inner product, and can efficiently distinguish memory patterns with similar directions but different norms. In terms of storage capacity breakthrough, the paper completes the theoretical proof via Theorem 4.7: when the norm bound of patterns is r_{max}, the model breaks through the performance limits of existing continuous Hopfield models and achieves a double-exponential improvement in storage capacity.

- Experimental Validation
This paper completes the performance validation of the model through multiple sets of experiments. In pattern completion experiments, the paper verifies on synthetic datasets and the dimension-reduced MNIST dataset that, under extremely low-dimensional conditions, the recall accuracy of the proposed hyperbolic model significantly outperforms the state-of-the-art MHN and DAM models. In downstream task validation, the paper encapsulates this mechanism as an attention layer that requires no additional hyperbolic parameters, and verifies its stable performance gains in low-dimensional latent spaces on tasks including MNIST image classification and multiple instance learning (MIL).

**Compliance With Llm Reviewing Policy:**

Affirmed.

**Key Questions For Authors:**

Regarding the choice of experimental baselines, please explain why you deliberately avoided comparisons with existing hyperbolic models in the field, In the classification and multiple-instance learning (MIL) tasks, the paper only compares the proposed Karcher Flow attention layer with modern Hopfield networks (MHN) and Dense Associative Memory (DAM) in Euclidean space. Given that this paper aims to establish the superiority of hyperbolic geometry in continuous memory models, why was there no fair comparison with mainstream hyperbolic attention networks in the field?
Regarding the convergence and optimization stability of the algorithm, the theoretical support for Proposition 4.4 shows a clear gap. The paper claims that the (KFM) update rule can iteratively approximate the optimal estimator via Karcher flow. However, careful examination of the formulation reveals that the attention weight w_\mu(v) is not static, but a nonlinear function that evolves dynamically with the state v at each iteration. Performing Karcher flow updates on a manifold for such a dynamic-weighted objective results in a highly non-convex optimization landscape that is prone to local deadlock. Please provide rigorous mathematical derivations to prove the global or local convergence of this iterative procedure.

**Limitations:**

yes

**Strengths And Weaknesses:**

Strengths:
	The paper establish a mathematical connection between the exponentially distributed place field sizes observed in biology and δ-hyperbolic metric spaces, providing a theoretical framework for understanding how neural population activity induces tree-like metric structures.
	For the proposed hyperbolic associative memory model, it is theoretically proven that when the stored patterns have differing norms, the model achieves a double-exponential increase in storage capacity compared to modern Hopfield networks in Euclidean space.

Weakness:
	In the validation on image classification and multiple instance learning (MIL), the paper only compared the proposed KarcherFlow layer with traditional Euclidean Hopfield networks. Given that the core point of this paper is the advantage of hyperbolic geometry in attention/memory mechanisms, why was no comparison made with existing mainstream hyperbolic attention networks in the field?
	In the CIFAR-10 pattern completion experiment shown in Figure 2, the proposed hyperbolic model completely collapses in performance, falling far short even of the DAM model. The paper only dismisses this issue by stating that “the dimensionality-reduced CIFAR-10 may lie in a regime where hyperbolic advantages cannot manifest”. If the hyperbolic model fails to handle even moderately complex natural image distributions, its theoretically derived “exponential capacity advantage” becomes meaningless in practical tasks.
	Section 4.4 of the paper claims that the Karcher flow update rule can iteratively approximate the optimal Fréchet mean estimator. However, the weight w_\mu(v) in the paper’s formulation is not a fixed value, but a nonlinear Softmax function that varies dynamically with the current iteration state v. Performing Karcher flow on a Riemannian manifold with such dynamic weights leads to a highly non-convex optimization landscape. The paper provides no mathematical proof whatsoever regarding the convergence or global optimality of this algorithm.

---

> ### Author Rebuttal · Authors · 2026-03-30
>
> **Weakness 1**: The validation experiments (image classification and MIL) only compared the proposed KarcherFlow layer against traditional Euclidean Hopfield networks...
>
> > The reason for this comparison is because in our derivation, KFM and MHN only differ in their underlying geometry (both are optimal-estiamtors under L2). Thus, with this comparison, we are able to observe how geometry could impact memory retrieval. To make the experiment more comprehensive, we added two other baselines [1,2] for classification and MIL. The results can be found in [**this link**](https://imgur.com/a/lQhZw9z). We can see that [2] performs the worst and has the highest variance. KF-attention and Hyperbolic attention networks ([1]) are comparable in most settings with KF-attention being slightly better on MNIST, and [1] slightly better on MIL.
>
> - [1] Hyperbolic attention networks (HAN) [Gulcehre et al., ICLR 2019]:
>
> - [2] Hyperbolic neural networks ++ (HNN++) [Shimizu, et al, ICLR 2021].
>
> **Weakness 3 and cifar10 performance**: (a) In the CIFAR-10 pattern completion experiment (Figure 2), the proposed hyperbolic model completely collapses in performance, falling far short even of the DAM model. (b) If the hyperbolic model cannot handle even moderately complex natural image distributions, the theoretically derived "exponential capacity advantage" loses practical significance.
>
> > We thank the reviewer for highlighting this. Our original hypothesis came from Table 3 in the appendix, which describes regimes where KFM memory storage could collapse. After carefully examining the reviewers' feedback, we provide an alternative explanation and additional simulation results.
>
> > Our experiments applied PCA as a preprocessing step. However, standard PCA respects the Euclidean metric rather than the hyperbolic metric [1], and may therefore not be suitable for KFM or non-Euclidean models in general. We thus perform pattern completion without PCA on CIFAR10. The results are shown [**here**](https://imgur.com/a/FrWIJ3v). Without PCA, KFM outperforms all baselines, likely because hyperbolic geometry is more robust against the curse of dimensionality, which is especially beneficial in high data dimensions (3072).
>
> [1] HoroPCA: Hyperbolic Dimensionality-reduction via Horospherical Projections (Chami et al. ICML 2021)
>
> **Weakness 4 + Question on KFM convergence**: The weight $w_{\mu}(v)$ in the formulation is not fixed...
>
> > We thank the reviewer for pointing this out. Indeed, our update is slightly different from the standard Karcher-flow algorithm. Therefore, the Frechet mean fixed point could differ from the KFM fixed point. However, we would like to highlight that Theorem 4.7 still guarantees high capacity of KFM.
>
> > Interestingly, the reviewer’s feedback suggests a new update rule to KFM by fixing the $w_{\mu} = w_{\mu}(v^{(0)}$. This allows KFM to converge to Frechet mean and also reduces repeated computation of the softmax weights. Based on this feedback, we believe our Remark 4.5 is incorrect. We have revised Remark 4.5 to the following:
>
> *One can consider the above proposition as the non-Euclidean version of Proposition 2.1. However, unlike Proposition 2.1, our tractable update rule (KFM) is not the exact optimal estimator. To converge to the optimal Frechet mean estimator, one can consider a fixed weight throughout the iterative updates. Given some initial state $v^{(0)}$, we set $w_{\mu}(v) = w_{\mu}(v^{(0)}).$ This reduces the computational cost for $w_{\mu}(v)$. However, in our later analysis, both update rules have the same order of stable fixed points under pattern separation.*
>
> ___
> Thank you for your time and valuable feedback. Please let us know if there are any other aspects of our work that you would like us to clarify. We look forward to further feedback and discussion!

---

### Official Review · Reviewer_wKxh · 2026-03-12

**Soundness:** 3
**Presentation:** 3
**Significance:** 3
**Originality:** 4
**Overall Recommendation:** 5
**Confidence:** 3

**Summary:**

The base idea of the babe is to that neural populations may naturally form hyperbolic representations, and decoding in that geometry leads to more efficient associative memory and inference. To this extent they show how neural tuning curves can induce hyperbolic geometry, they then establish a connection between neural decoding and associative memory, showing that the update rule of modern Hopfield networks corresponds to computing the Bayes-optimal minimum mean squared error (MMSE) estimator for decoding stimuli from neural activity.and based on these ideas they propose a new hyperbolic associative memory model with higher capacity than modern Hopfield networks. They use this model to study pattern completion and machine learning tasks, demonstrating through theoretical analysis and simulations that hyperbolic memory representations can improve recall capacity and performance, particularly in low-dimensional representation regimes, suggesting that neural systems and artificial models may benefit from representing information in hyperbolic latent spaces.

**Compliance With Llm Reviewing Policy:**

Affirmed.

**Key Questions For Authors:**

- How realistic is the assumption that place field widths follow an exponential distribution to obtain hyperbolic geometry?
- Can you provide a comparison of computational analysis for the hyperbolic model you have developed vs the existing ones?
- How sensitive are the Hopfield networks you proposed to correlations and similarities in data?
- The experiments are done on relatively small datasets and low-dimensional settings. Would the proposed approach still provide improvements when used in larger machine learning models or real-world tasks?

**Limitations:**

- Compute might be a limitation, because this doesn't really seem scalable.
- I think a sensitivity analysis on how strong is the models retrieval when very similar things are stored would be needed to say more about the sensitivity of the storage.

**Strengths And Weaknesses:**

- The work provides a clear theoretical framework linking neural population coding, geometric representation, and associative memory, which is conceptually strong and unifies ideas from computational neuroscience and modern Hopfield networks.
- The paper establishes a novel theoretical connection showing that modern Hopfield network updates correspond to the Bayes-optimal MMSE estimator for neural decoding.
- Several assumptions underlying the theoretical analysis are relatively strong, such as the exponential distribution of place-field widths, very large neuron populations, and specific similarity metrics, which may limit how generally the hyperbolic geometry result applies to real neural systems.
- The empirical validation is relatively limited to smaller datasets and doesnt support or bring about much results on larger datasets

---

> ### Author Rebuttal · Authors · 2026-03-30
>
> **Weakness 1 and Question 1**: Several assumptions underlying the theoretical analysis are relatively strong…
> > Assumptions in the paper are well-motivated by either the experimental evidence or neuroscience conventions.
> The exponential distribution of place-field widths is discovered in [1, 2]. The system we are studying is the CA1 region of the hippocampus, where in an adult rat, it has roughly 300k to 400k neurons [2], which justifies $N$ being sufficiently large. The design of our similarity metrics are inspired by Representational similarity analysis (RSA) [3], a common approach of studying neural population geometry, which generates numerous empirical results in neurobiology. In RSA, the most common metric in fact follows our construction where one first applies inner product between population activities, and then applies some continuous function. Thus, the metrics experimentalists use in neurobiology are mostly captured by our formulation.
>
> [1] Rich, P. D., Liaw, H.-P., and Lee, A. K. Large environments reveal the statistical structure governing hippocampal representations. Science (2014).
>
> [2] Zhang, H., Rich, P. D., Lee, A. K., and Sharpee, T. O. Hippocampal spatial representations exhibit a hyperbolic geometry that expands with experience. Nature Neuroscience (2023).
>
> [3] Kriegeskorte, N., Mur, M., Bandettini, P. A. Representational similarity analysis - connecting the branches of systems neuroscience, Frontiers in Systems Neuroscience (2008).
>
> **Weakness 2, Question 4 & limitation 1**: (W) The empirical validation is relatively limited to smaller datasets and doesnt support or bring about much results on larger datasets. (Q) The experiments are done on relatively small datasets and low-dimensional settings. Would the proposed approach still provide improvements when used in larger machine learning models or real-world tasks?
> > We thank the reviewer for raising this important point. Computational scalability is indeed a key bottleneck for hyperbolic networks, stemming from operations such as the exponential and logarithmic maps, which are costlier than their Euclidean counterparts. While recent works have proposed methods to mitigate this [1], hardware- and algorithm-level optimizations remain an open challenge for the broader non-Euclidean ML community. As our work focuses on the modeling and theoretical contributions of KFM, we acknowledge this as a limitation and defer a full treatment to future work. Since our KF-attention module is fully compatible with Euclidean optimizers (e.g. Adam), a potential direction is to directly apply KF-attention to existing deep network architectures. We have added a dedicated discussion in the limitations section accordingly.
>
> [1] Lorentzian Residual Neural Networks (KDD 2025, He et al.)
>
> **Question 2**: Can you provide a comparison of computational analysis for the hyperbolic model you have developed vs the existing ones?
>
> > We compare KF-attention, HAN [1] and HNN++ [2]. All three methods are quadratic in sequence length $N$. The subtle differences are as follows. KF-Attention computes the inverse exponential map $\operatorname{Exp}^{-1}_{q_i}(\cdot)$ at each query point $q_i$​, which requires $N^2$ evaluations of $\operatorname{arccosh}$, which is the primary computational bottleneck. HAN avoids this as its aggregation is a closed-form Einstein midpoint requiring only $N$ precomputed Lorentz parameters, reused across all queries. The trade-off is that KF-Attention performs geometrically faithful aggregation at each query point, while HAN uses a one-step approximation. HNN++ avoids $\operatorname{arccosh}$ as well, but its parameters live in hyperbolic space, requiring Riemannian optimizers. KF-Attention and HAN are fully compatible with standard Euclidean optimizers (e.g. AdamW), which is a practical advantage since Riemannian optimization introduces significant implementation complexity and computational overhead.
>
> [1] Hyperbolic attention networks [Gulcehre et al., ICLR 2019]
>
> [2] Hyperbolic neural networks ++[Shimizu, et al, ICLR 2021]
>
> **Question 3 & limitation 2**: How sensitive are the Hopfield networks you proposed to correlations and similarities in data?
>
> > Here we conduct an analysis on memory retrieval **within classes** in the CIFAR10 dataset. This increases similarities between stored memory patterns. We slightly modify the preprocessing and perform pattern completion on raw flattened images without PCA. The results can be found in the links below. We are able to see that KFM outperforms other baselines in every setting. (We set $\beta=1.0$ for all runs).
>
> CIFAR 10 ([full](https://imgur.com/a/FrWIJ3v), [cat](https://imgur.com/a/HarUNuy), [dog](https://imgur.com/a/d7BLWzU), [frog](https://imgur.com/a/hv4g1fp))
> ___
> We thank the reviewer for the positive evaluation of our work. Please do not hesitate to let us know if there are any other aspects of our work that you would like us to clarify.

---

> > ### Author Rebuttal · Reviewer_wKxh · 2026-04-04
> >
> > Thank you for your responses. I am satisfied with the reasoning provided, but I would reserve the same scores.

---

### Official Review · Reviewer_TNbw · 2026-03-12

**Soundness:** 3
**Presentation:** 2
**Significance:** 4
**Originality:** 3
**Overall Recommendation:** 5
**Confidence:** 3

**Summary:**

This paper studies stimuli encoded with Gaussian place fields with place field kernels having exponentially distributed widths, and shows that—as the number of neurons and the domain size increases—the geometry appears to become “tree-like”. This is indicative of a hyperbolic geometry. The paper also shows a connection between modern Hopfield networks and neural place-field codes and, inspired by this and by the hyperbolic shape of place field geometry, suggests a new modern Hopfield style method that operates in a hyperbolic space. It is shown, theoretically and empirically, that this hyperbolic modern Hopfield network has greater capacity than standard versions.

**Compliance With Llm Reviewing Policy:**

Affirmed.

**Final Justification:**

Modulo my uncertainty based on not having read much of the appendix (as mentioned in the original review), I appreciate the paper and will recommend it for acceptance. I am not giving it full points, however, because of the originality weakness mentioned in my original review, and because, given that I cannot review the full updated manuscript, I would tend to reserve full points for papers that have a higher degree of clarity of presentation upon first submission.

**Key Questions For Authors:**

Questions
1. page 2: “we discretize the stimulus space into $M$ grid points … and assume a uniform prior”. What happens if you do not assume a flat prior?
2. line 152, right column: “and cyclic permutations” => what does this mean in the context?
3. figure 2: What does this mean: “We suspect that CIFAR10 after dimension reduction, falls into the regime where the superior scaling does not show under hyperbolic geometry.”
4. line 358, right col: why is the likelihood for $\mu$ supposed to be small? Is this not a typo?
5. line 1123: We require the signal $K_max$ to be at least of order $L^{-2}$. I didn’t see this assumption in the theorem statement. Did I miss it or was it not there? What does this assumption correspond with?

**I have added potential minor errors that I found below, as I was not sure where else to put them in the review:**

Missing definitions:
- page 1: what is meant by “statistically hyperbolic”? This could perhaps use a 1-sentence definition.
- line 133-134: $\phi$ is introduced very quickly and it is difficult to follow what it corresponds to in the standard modern hopfield formulation. Is it possible to elaborate? Why is it applied to the stimulus and the spiking?
- line 134-35: $v$ is introduced with no definition
- line 145-46: $\mathcal{V}$ not defined
- line 171, right column: $d_\mathcal{M}$ is not defined
- line 245-46: what are $s_{(1)}$ and $s_{(2)}$?
- paragraph at 229, right side: what is the difference between $\psi_i$ here and $\psi_2$ later in the paragraph? Are there conditions on these?
- line 283: what is $\mathcal{N}$?
- line 288, right side: what is the subscript on $\mathrm{Exp}$ denoting?
- assumption 4.6: $v$ is not defined; what is the superscript on $\Delta$?

Minor notes:
- line 100: “the case of single-field” – missing an ‘a’
- line 105/106, left column: grammar issues
- line 93/94, right column: “which is normally distribution asymptotically” => which is normally distributed, asymptotically.
- line 165: grammar issues; missing ‘is’, ‘a’.
- line 183, right column: “the semi-metric space underlies it” is missing ‘that’
- line 191-92, right col: grammar issues
- line 218, right col: suggest writing “single place field” instead of “single-field”, to be clear what is being discussed
- Th 4.1: “Furthermore, $\delta$ non-trivial as” => I think you mean “Furthermore, for…”
- line 258: missing ‘a’
- line 267-272: several grammar issues
- line 221, right col: grammar
- line 234, right col: “Given an observed spikes” => “Given observed spikes”
- line 239, right col: “under squared” => “under a squared”
- line 296-97: grammar
- line 301: “In contrast, Euclidean inner product in general does not represent Euclidean distance” => In contrast, the Euclidean inner product in general does not represent hyperbolic(?) distance
- fig 2: “in 'b', colours R1-R6 in legend do not match plot”
- fig 2: missing ‘b’ label in figure
- line 418, right col: “dimension” => perspective
- line 1047: appears to be missing squares
- line 1050: appears to be missing brackets and addition symbol
- line 1077: appears to be missing minus sign before $ln$
- line 1085: “decompose” => decomposing
- line 1131: “solve” => “solving”

**Limitations:**

Given the implications for neuroscience and, via the modern Hopfield connection to transformers, to AI, there is clearly potential for societal impacts here, but these issues are not discussed in any depth. For example, this work could lead to novel LLM architectures—which could improve chat bots and aid in writing and web queries—but also, through increased AI adoption, further scale up the damaging aspects of LLMs—like significant C02 emissions, increased wealth concentration and power in the hands of “big tech”, and job loss in many sectors. Engaging with such positives and negatives for AI, along with neuroscience, would be quite useful. As a starting point for the AI impacts, see the work of authors like Alex Hanna, Emily Bender, and Kate Crawford.

**Strengths And Weaknesses:**

I go through the soundness, presentation, significance, and originality, below. I label (in brackets) whether I consider each to be a strength or weakness).

Soundness (uncertain, as I have not had time to check the supplementary):
- Beyond typos, the paper seems sound. Although I cannot confirm this as I have not had time to extensively review the supplementary section. **Because I did not check the proofs I have marked my review as having a low degree of certainty.** I have rated the paper a 3 for soundness, assuming that the proofs are correct.


Presentation (strength and weakness)
- strength: overall organization of the paper is nice and logical
- weakness: there are many quantities that are not properly defined (see questions section). This makes it difficult to follow.
- weakness: there are many grammar issues

I don’t think these weaknesses should be too difficult to fix!

Significance (strength)
- I have not yet seen a theoretical argument for hyperbolic geometries in place cells, which makes this significant
- The hyperbolic version of the modern Hopfield network seems significant

Originality (strength)
- Both points regarding significance make the paper sufficiently original, in my view. I have rated the paper a 3 for originality because of the existing empirical work showing that neural place field geometry seems to be hyperbolic

**I have rated the paper as a weak reject primarily because of the issues of presentation and the impact assessment (see limitation section). I think these issues should be easily resolved, and I will increase my score if they are.**

---

> ### Author Rebuttal · Authors · 2026-03-30
>
> We thank the reviewer for their insightful feedback and attention to detail. We have revised our manuscript accordingly, addressing the noted typos and missing definitions. We look forward to further discussion.
>
> **Question 1**: what if we do not assume a flat prior?
> > The prior term here indicates how animals sample (experience) the environment. We assume a flat prior to simplify our analysis. The direct impact of not assuming a flat prior is equation (2.6) will contain a bias term $\log p(i)$, i.e., $s^*(n)=\sum_{i}\operatorname{softmax}_{i}(h_i(n)+\log p(i))$ (here we use $i$ instead of $\mu$ for latex rendering reasons. Studying non-flat priors is an interesting future direction. For instance, animals tend to visit environment boundaries more frequently. We have added a brief discussion of this in the appendix.
>
> **Question 2**: cyclic permutations
> > Cyclic permutations of $(x,y,z)$ refers to the three rotations: $(x,y,z), (y,z,x), (z,x,y)$. In Def. 3.1, a $\delta$-thin triangle requires the thinness condition to hold under all three of these cyclic relabelings of its vertices.
>
> **Question 3**: We suspect that CIFAR10, after dimension reduction, falls into…
> > This refers to Table 3 (page 23), which identifies a regime where the model collapses. However, based on other reviewers' feedback, we conduct a new simulation on CIFAR10 without PCA as PCA might not be suitable for non-Euclidean models. We can see KFM indeed outperforms other baselines. (See [**this link**](https://imgur.com/a/FrWIJ3v)) We suspect this is because PCA is not suitable under hyperbolic-based models.
>
> **Question 4**: line 358, right col: why is the likelihood for mu supposed to be small?
> > It is a typo, the correct threshold should be $p( \mathbf{v} \mid \mu) >= 0.99$.
>
>
>
> **Question 5**: We require the signal $K_{max}$ to be at least of order ….
> > It is a condition which we need to make the theorem hold. In our proof (Line 1153, page 21), we further show that this condition is satisfied with sufficiently large $N$ by setting $N > exp\left( \frac{L}{2\beta \sqrt{\ln L}} \right) $. We have also revised the statement of theorem 4.1 to be:
> ___
>
> Let $(x,y,z,w) \subset \mathcal{S}$ to be any quadruple uniformly sampled by index.
> For any $\eta > 0, N > \exp\left( \tfrac{L}{2\beta \sqrt{2 \ln L}} \right)$, there exists a constant $\delta(\beta, \rho)$ such that as the center density $\rho = \tfrac{N}{\text{Vol}(\mathcal S)} \rightarrow \infty$, $\textbf{Pr}[ \Delta(x,y,z,w) > 2 \delta ]<\eta.$ Furthermore, $\delta$ is non-trivial as $\lim_{L\rightarrow \infty} \frac{\delta}{L} = 0$.
>
> ___
>
>
> **Question regarding definitions of $\psi, \phi$**
>
> > We apologize for the confusion. We update our notation for clarity. Let $\psi^E: \mathbb{R}^N \rightarrow \mathbb{R}^d, \psi^H: \mathbb{R}^N \rightarrow \mathbb{H}^d$ denote the non-linear maps connecting the tuning curve encoder to MHN and KFM, respectively, both taking $\lambda(s)$ as input. This decoupling avoids imposing a strict biological constraint between encoder and decoder, while ensuring compatibility with the Boltzmann form assumed in Propositions 2.1 and 4.4. Those mappings are not seen in the typical MHN works as existing works do not consider the case where inputs are generative by some encoder (tuning curve).
>
> **Question regarding other definitions**
> > We apologize for the confusion and have added a link to the notation table (Table 2) in the main text. Brief clarifications can be found in [**this link**](https://imgur.com/a/Eg028xs):
>
> **Limitations**
> > We thank the reviewer for highlighting the potential societal impact. While our work primarily focuses on neuro-inspired AI, it could have broader implications for society, particularly through its connection to transformers and modern LLMs. We have added the following to our manuscript:
>
> > (Hanna & Bender) The reliance on massive computational resources concentrates power within a few large companies (e.g., Google). Our method introduces an additional perspective, hyperbolic geometry, while keeping the overall model size unchanged. Despite this added complexity, the required computational resources remain comparable, yet the model demonstrates superior memory capacity. This improvement in efficiency could lower the barrier to entry, enabling smaller companies and research labs to develop competitive models, thereby helping to reduce power concentration in AI research.
>
> > (Crawford) Current LLMs largely depend on scaling laws to increase intelligence, which leads to significant carbon emissions. While our model does not fully resolve this issue, it provides a promising, under-explored approach for energy-efficient AI. As shown in Theorem 4.7, our model achieves double-exponential scaling in memory capacity, allowing efficient performance even in strictly low-dimensional settings. This offers a theoretical foundation for designing more energy-conscious architectures.
>
> ___
>
> Thanks again for the insightful feedback!

---

> > ### Author Rebuttal · Reviewer_TNbw · 2026-04-02
> >
> > Thanks for the clarifications! I have a few follow-up questions and comments:
> > - Regarding **Q2**: I might suggest adding a bit more text to clarify. E.g. "and cyclic permutations" => "and that this holds under cyclic permutations of the triangle"
> > - On $\psi$ and $\phi$: why introduce these embeddings and not simply define the dynamics on the space of neural spike counts $n$? Is it because you wish the dynamic space to be continuous? One extra minor notational point related to $\psi$: it seems an abuse of notation to write it mapping from both spike count space and stimulus space on lines 134-135
> > - Thank you for updating the points on broader impact! My one other comment here is that it could be worth providing examples of potential negative impacts as well, for completeness. For example, potential energy efficiency benefits might lead to increased use of ML models in small devices for surveillance and weapons applications, or "[Jevons Paradox](https://en.wikipedia.org/wiki/Jevons_paradox)" might paradoxically lead to an increase in energy use through efficiency gains.
> > - Lastly, I noticed an extra typo in the appendix. In the Eq. on line 1025 I believe $s'$ should be $s_i$

---

> > > ### Author Response · Authors · 2026-04-04
> > >
> > > Dear Reviewer TNbw,
> > >
> > >
> > > Thank you for your valuable feedback and questions!
> > >
> > > ___
> > >
> > > - Regarding Q2, we agree that adding more text would help clarify the definition. We have modified our definition accordingly in the manuscript.
> > > - Regarding $\psi, \phi$: Yes, one key purpose of the mappings is to maintain continuous dynamics. This is necessary to establish our Karcher-flow analysis as we need the state space to be a smooth Riemannian manifold to define tangent space and the exponential map. We now restate both (a) line 130-135 and (b) line 230-235 right-col below. (lines 283-285 are also modified accordingly)
> > >
> > > > (a) Modern Hopfield networks \cite{ramsauer2020hopfield, krotov2020large} (MHNs) are associative memory models defined over continuous state spaces. Let $\psi^E_1 : \mathcal{N} \rightarrow \mathbb{R}^d$ be an embedding map for the query. $\psi^E_2: \mathbb{R}^N \rightarrow \mathbb{R}^d$. We define the network state as $v = \psi^E_1(n)$ and stored memory patterns $\xi_\mu = \psi^E_2(\lambda(s_\mu))$.
> > >
> > >
> > > > (b) Given a set of preferred stimuli $s_{\mu}$, $\mu \in [M]$, let $\psi^H_1: \mathcal{N} \rightarrow \mathbb{H}^d$, $\psi^H_2: \mathbb{R}^N \rightarrow \mathbb{H}^d$ such that $\xi_{\mu} = \psi^H_2(\lambda(s_\mu))$, $\mathbf{v}= \psi^H_1(n)$.
> > >
> > > To summarize our modification, we first introduce the superscript $E, H$ to specify the maps for Euclidean and hyperbolic models. Next, we separate the maps for query (from spikes) and memory patterns with subscript $1,2$.
> > >
> > > - Regarding $s^\prime$, it actually meant to be $s^\prime$ instead of $s_i$. Since our goal here is to show that neural population codes ($\lambda(s)$) are hyperbolic embeddings of the stimulus space $\mathcal{S}$ we need to define a distance function  $d(s, s^\prime)$ between any two points in the stimulus space $\mathcal{S}$ instead of between just the preferred stimuli. Finally, this definition allows us to later see that given any two inputs $s, s^\prime$, the function $d(s, s^\prime)$ is tree-like (hyperbolic).
> > >
> > > - We thank the reviewer for this thoughtful suggestion. We have added the following to the broader impact statement: "However, we acknowledge potential negative consequences. Efficiency gains in memory and representation may enable deployment of ML models in resource-constrained devices, which could facilitate surveillance or autonomous weapons systems. Furthermore, consistent with Jevons Paradox, improvements in computational efficiency may paradoxically increase overall energy consumption by lowering the barrier to large-scale deployment."
> > >
> > > ____
> > > We thank the reviewer for this suggestion! We hope our revisions have addressed your concerns, and would greatly appreciate it if you would consider raising your score.

---

### Decision · Program_Chairs · 2026-04-30

**Decision:**

Accept (regular)

**Comment:**

This study considers a population of neurons of Gaussian tunings whose tuning widths are exponentially distributed. It found this setting is sufficient to produce a hyperbolic latent space manifold. It also shows a connection between neural population decoding and modern Hopfield networks. The network with the hyperbolic structure exhibits higher capacity, demonstrated by the experiments on MNIST and CIFAR10 datasets.

Most reviewers appreciate the theoretical soundness of the study. The rebuttal addressed most of the reviewers’ concerns. So I recommend accepting this paper. Please revise and proofread the manuscript carefully to address the writing issues as pointed out by the reviewers.